# Gender and racial diversity socialization in science

Weihua Li [1,2,3,4,5,6], Hongwei Zheng [7] ✉, Jennie E. Brand[8] & Aaron Clauset [9,10,11] ✉

Scientific collaboration networks are a form of unequally distributed social capital that shapes both researcher job placement and long-term research productivity and prominence. However, the role of collaboration networks in shaping the gender and racial diversity of the scientific workforce remains unclear. Here we propose a computational null model to investigate the degree to which early-career scientific collaborators with representationally diverse cohorts of scholars are associated with forming or participating in more diverse research groups as established researchers. When testing this hypothesis using two large-scale, longitudinal datasets on scientific collaborations, we find that the gender and racial diversity in a researcher's early-career collaboration environment is strongly associated with the diversity of their collaborators in their established period. This diversity-association effect is particularly prominent for men. Coupled with gender and racial homophily between advisors and advisees, collaborator diversity represents a generational effect that partly explains why changes in representation within the scientific workforce tend to happen very slowly.

The scientific workforce in the United States has never been representative of the general US population. For example, men and White researchers currently comprise a disproportionate majority of tenure-track US faculty and also have lower attrition rates compared with women[1–3], while Black and Latine researchers are far below demographic proportionality[4], which is often most severe in science, technology, engineering and mathematics (STEM) fields[5,6]. Although gender and racial diversity in the scientific workforce have increased over time, the rates of change have often been remarkably slow[7]. For example, the proportion of women among newly hired tenure-track faculty at US universities did not increase substantially for most fields between 2011 and 2020[1]. Despite many decades of study, we still lack a complete accounting of the distinct social processes that influence academic workforce diversity.

Contrasting the slow pace of diversification are the well-documented utilitarian benefits of scientific workforce diversity: in creative activities such as scientific discovery, workforce diversity produces more innovative ideas[8–10]. As a result, the lack of diversity in the US academic workforce probably limits the range and rate of scientific and technological discoveries. For instance, the dearth of women inventors implies a genuine loss of innovation for society[11,12]. Comparable effects have also been identified for Black inventors[13] and inventors from less privileged socioeconomic backgrounds[10].

However, interest in understanding the persistently low demographic diversity of US academic faculty stretches back at least to the 1990s[14]. This line of research often focuses on 'pathway' analyses, in which disparities at the individual level earlier in an educational sequence or career progression tend to limit the possible diversity in subsequent stages[15,16]. In most STEM fields, women remain under-represented in graduate applications and completion, which limits the number of women that are later eligible for permanent

[1]LMIB, NLSDE, BDBC, and School of Artificial Intelligence, Beihang University, Beijing, China. [2]Hangzhou International Innovation Institute, Beihang University, Hangzhou, China. [3]Department of Advanced Interdisciplinary Research, Pengcheng Laboratory, Shenzhen, China. [4]Zhongguancun Laboratory, Beijing, China. [5]Qianyuan Laboratory, Hangzhou, China. [6]Beijing Advanced Innovation Center for Future Blockchain and Privacy Computing, Beihang University, Beijing, China. [7]Beijing Academy of Blockchain and Edge Computing, Beijing, China. [8]Department of Sociology, University of California, Los Angeles, Los Angeles, CA, USA. [9]Department of Computer Science, University of Colorado, Boulder, CO, USA. [10]BioFrontiers Institute, University of Colorado, Boulder, CO, USA. [11]Santa Fe Institute, Santa Fe, NM, USA. ✉e-mail: hwzheng@pku.edu.cn; aaron.clauset@colorado.edu

scholarly positions[17,18]. Biases in faculty hiring can further decrease women's representation[3,19], and even when they are hired, their retention rates are substantially lower, via work–life incongruencies such as the unequal impact of parenthood[20,21], workplace climates that favor men[22] and double standards in promotion evaluations[23].

Racial minority researchers also experience substantial cultural, climate and socioeconomic barriers. Black and Latine youth are less likely to enter STEM majors compared with their White peers[24], experience higher dropout rates in college[25] and have lower rates of graduate enrollment to become doctoral degree recipients[26]. They also face notable disparities in faculty hiring, promotion and retention, which further impede their representation in academia[27]. Notably missing from most pipeline analyses are considerations of endogenous processes such as network effects, in which interactions among individuals can influence downstream outcomes, both positively or negatively.

A key example of such a process is academic socialization, which plays a crucial role in transmitting academic values and norms[28,29], development of scientific skills[30,31], shaping social network dynamics and allocation of social capital[32,33], and learning to be an independent scholar[34]. Attitudes and beliefs about work and ethics tend to stabilize in the early career of academics and are unlikely to change later in a career[35,36]. In mentor relationships, advisees tend to adopt from their advisors' practices that facilitate their integration and success within their scientific community[37], and emulate the social behavior and career choices of their advisors[38]. Collaboration networks provide another mechanism by which early-career researchers can be influenced by senior scholars, such that connections to elite senior researchers can increase the exposure, prominence and career prospects of early-career researchers[39–41]. Hence, a clearer understanding of how different experiences of academic socialization influence downstream choices by researchers would connect evidence of the scientific and social advantages of diverse teams[9,10,42] with potential policy or cultural interventions to accelerate the benefits to science and society.

Here we investigate another aspect of academic socialization: whether and to what degree the representational diversity of a researcher's early-career environment correlates with a tendency to subsequently construct relatively diverse collaboration networks as an established researcher. This 'diversity association' effect hypothesizes that academic socialization extends beyond the adoption or transmission of the practicalities of successful research to also encompass attitudes and preferences around the social components of being a scholar, and in particular, the choices a researcher makes about the composition of their collaborative environment. This hypothesis is grounded in the empirical evidence for the broad influence that academic socialization can have on researcher behavior, and connects early experiences with representational diversity of social identities as a doctoral trainee to preferences for diversity as a doctoral supervisor, in contrast to simple expectations of homophily (Fig. 1a). We conceptualize this process in the post-functionalist sense[29] as a form of 'diversity socialization', an idea we develop subsequently.

Empirically, diversity association reproduces itself, such that advisees trained within more sociodemographically diverse groups will be more likely to lead similarly diverse environments when they themselves become advisors. A similar diversity-association effect has been observed in non-academic situations, where White children who had more Black peers of the same gender in their grade were more likely to live in districts with more Black residents when they become adults[43]. Academic socialization may be particularly relevant for under-represented cohorts of students, as research suggests that mentoring is often inadequate for graduate students of color[30].

However, measuring the effects of diversity association is complicated by demographic trends toward greater gender, racial and ethnic diversity in many fields[44,45], which should naturally increase the diversity of advisee groups, regardless of how a supervising researcher was trained. Variation in demographic diversity across subfields and

institutions can also skew estimates of the diversity-association effect, such that researchers in high-diversity subfields and institutions should naturally have more diverse advisees than those at less diverse institutions. For example, a medical research group of seven students with three women is common, as gender representation in the medical workforce is now close to parity. In contrast, a computer science research group with six members and two women may be considered gender diverse, because women's representation in computer science has grown over the past several decades to be around 20–30%. Hence, the diversity-association effect must be measured relative to a null expectation that accounts for observed subfield and institution demographic diversity at a particular time. In this study, we measure the diversity of a group relative to the scientific workforce as a whole, rather than relative to any particular individual. A group is diverse if it contains relatively more minority members than we would expect at random, given their representation at the workforce level. In this way, an individual can be a part of a diverse group regardless of whether they themselves have a minority or a majority social identity.

We apply such a subfield- and trend-adjusted null model to two broad, multidisciplinary datasets of observed advisor–advisee relationships and collaboration networks in STEM subfields. The first dataset comprises start and end years for 339,744 advisor–advisee pairs spanning STEM fields, social sciences and humanities[46], from which we select 17,917 established researchers (Methods). The second dataset is derived from records of 30.6 million research articles in the Microsoft Academic Graph (MAG) database from 1950 to 2021, covering publications in natural sciences, engineering, mathematics and social sciences[47] (Methods). Using name-based gender and racial assignment, we select 562,494 established researchers for the gender analysis and 855,526 researchers for the racial analysis (Supplementary Note 1).

## Results

### Gender diversity association

In mentor relationships, we define a research group to comprise all advisees mentored by the same advisor during the training period of a particular advisee, defined as the interval between the start and end years (inclusive) of an advisee's stay within the group. Among the established researchers we selected, 86.3% of men trained in research groups that were majority man, while only 36.4% of women trained in man-majority groups; 66.4% of focal researchers were trained in small research groups with 5 or fewer advisees, while 15.0% received training in large research groups with more than 10 advisees (Fig. 1b).

Among established researchers, we find that both men and women who trained in woman-majority groups were substantially more likely to advise women trainees when they became advisors themselves than researchers who trained in man-majority groups, regardless of the researcher's gender (Fig. 1c). Men who trained in woman-majority groups go on to advise groups that are on average 35.4% women, compared with on average 18.2% for men who trained in man-majority groups. In general, we find that women researchers are much more likely than men to have women advisees, indicating a basic level of gender homophily in the advisor–advisee tie formation process. However, women who trained in woman-majority groups go on to advise groups consisting of, on average, 50.3% women, compared with, on average, 34.3% for women who trained in man-majority groups.

In the collaboration networks, we retain researchers with at least 10 years between the first and most recent publication and at least 10 publications in total. To these, we apply a name-based gender-labeling algorithm to assign binary gender labels to the 562,494 authors that have junior collaborators in both the early-career and established periods (32.3% women and 67.7% men). We define the 3-year period (inclusive) that begins with a researcher's first publication as their training period and we define the period starting in their 6th year as their established-researcher period. Early-career researchers or co-authors are defined as scholars in their training period and

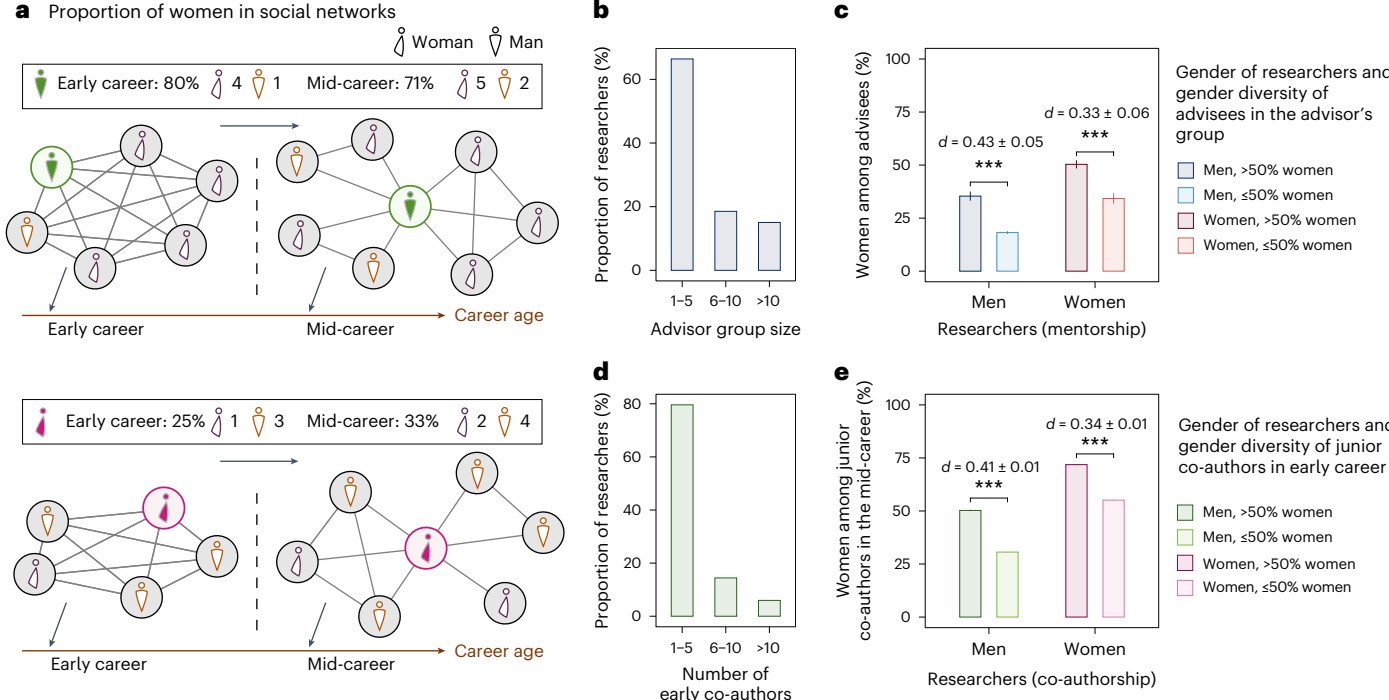

**Fig. 1 | Diversity-association effects in the social networks of researchers in the early-career and established periods. a**, An illustrative example of the gender diversity-association effect for two researchers during the early-career and established periods. The early social network of the red author comprises mostly women, and his network in the established period continues to be dominated by women. The green author interacted more often with men in their early career, and her connections in the established period are mostly men researchers as well. **b**, The group size distribution of advisors for researchers in the mentorship data.

**c**, The fraction of an advisor's advisees who are women as a function of advisor gender and the gender composition of their own early-career environment (*n* = 17, 917). **d**, Number of early junior co-authors of researchers in the collaboration network. **e**, The correlation between the proportion of women junior co-authors for researchers in the early-career and established periods (*n* = 562, 494). Bars represent mean values and error bars indicate 95% confidence intervals in **c** and **e**. We report effect sizes using the Cohen's *d* statistic and use two-sided *t*-tests for comparisons. ***$P < 0.001$.

established researchers are defined as those who have entered their established period. In both the early-career and the established periods, we analyze only the junior co-authors of each selected researcher in the co-authorship network. As such, for a given researcher, these distinctions ensure that established researchers are never counted as junior co-authors, and any observed career-wise diversity-association patterns must be caused by factors other than repeated collaboration with early co-authors. Among them, 79.6% collaborated with 5 or fewer junior co-authors in the early-career period, while 6.0% of researchers had more than 10 junior co-authors (Fig. 1d).

Among the selected established researchers, about 75.4% of men had a set of junior co-authors in their early career who were majority man. In contrast, only 55.4% of women's early-career junior collaborators were majority man, again reflecting a level of baseline gender homophily. We find that researchers with majority-woman early collaborators are much more likely to collaborate with women junior co-authors when they themselves become established researchers (Fig. 1e). This effect is particularly strong for men: those whose early co-authors were majority woman have, in their mid-career period, junior co-author collaboration groups that are, on average, 50.3% women, compared with 30.8% women for researchers whose early co-authors were majority man. For women, those with majority-woman early co-authors have on average 71.2% women junior co-authors in their established period, compared with 55.0% for researchers who had man-majority early co-authors. These patterns are consistent with our hypothesis about the gender diversity-association effect in academic careers, but do not account for the effects of demographic variability across subfields, institutions or time.

Simply observing a woman- or man-majority research group does not necessarily reflect a propensity of forming more or less

gender-diverse groups under a specific environment, because compositions are constrained by the demographics and structure of that environment at that time. To account for such effects, we introduce a null model to randomize the mixing patterns of social networks, preserving each individual's number of interactions, while controlling for structural constraints on associations from time, subfield, country and institutional prestige (Methods). The result is a kind of permutation test, in which we randomize each observed established–junior pair (either advisor–advisee pair or senior–junior co-author pair) among a set of structurally plausible alternatives, drawing uniformly from the set of all junior researchers in the same year, in the same subfield (MAG level 1) and country, and located at institutions in the same prestige category to the observed pair. In each null model instance, we randomly permute all mentor relationships or co-author pairs and we measure the expected demographic diversity of researchers by averaging it over 100 independent model instances. We then define researcher *i*'s group of advisees or co-authors to have a high percentage of women if the observed diversity exceeds the expected diversity under the model:

$$\rho_i > \langle \rho_i^{\text{null}} \rangle, \quad (1)$$

where $\rho_i$ is the observed proportion of women in *i*'s group and $\langle \rho_i^{\text{null}} \rangle$ is the average proportion of women expected under the null model.

Within the mentorship data, we find substantial evidence for a diversity-association effect. For men, 49.4% of those who were trained in groups with a high percentage of women, relative to the null model, themselves have a high percentage of women among their own advisees, compared with 34.9% for those who trained in groups without a high percentage of women (Fig. 2a). For women, 64.2% of those who were trained in groups with a high percentage of women go on to have a

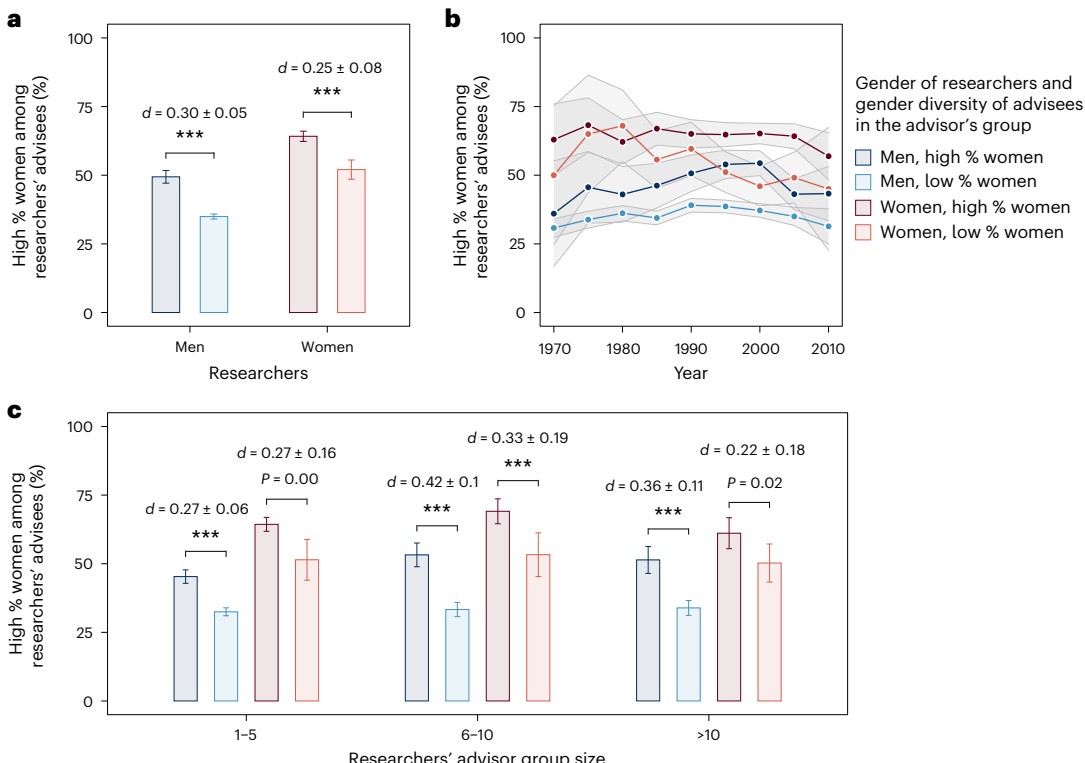

**Fig. 2 | Comparing gender diversity of advisors' groups and researchers' own groups.** The association effects of gender diversity relative to the null model using the mentorship data. **a**, The interplay between the gender diversity of advisees in the group of a researcher's advisor and that of his/her own group, for men and women. The cohort of advisees in a research group is defined to have a high proportion of women if the real percentage of women is greater than the baseline rate predicted by the null model. **b**, The gender diversity-association effects over time from 1970 to 2010, based on the researchers' first year as advisees. As data are less available for researcher groups in the early period, we merge data points for every 5 years instead of displaying yearly data. **c**, The interplay between the research group size of researchers' advisors and the gender diversity-association effects. Bars represent mean values and error bars indicate 95% confidence intervals in **a** and **c**. Lines denote mean values and shaded areas represent 95% confidence intervals in **b**. We report effect sizes using the Cohen's d statistic and use two-sided t-tests for comparisons. \*\*\*$P < 0.001$.

high percentage of women among their advisees, in contrast to 52.0% of those who trained without a high percentage of women. Furthermore, we find that this substantial effect size is relatively stable over time for all gendered groups of researchers (Fig. 2b).

In addition, we analyze the diversity-association effect with different group sizes (Fig. 2c). For those whose advisors managed research groups of 6 to 10 advisees, 53.2% of men who were trained in groups with high percentages of women went on to mentor a high percentage of women among their own advisees, compared with 33.3% for those whose advisors' groups had a low percentage of women. Similarly, 69.1% of women who were trained in groups with a high percentage of women went on to mentor a high percentage of women among their own advisees, compared with 53.3% for those whose mentors' groups had a low percentage of women. We find marginally smaller, but still statistically significant effect sizes for researchers who trained in groups with five or fewer advisees.

Within the co-authorship data, we find similar evidence for diversity-association effects. For men, 63.1% of those whose early-career junior co-authors were composed of a high percentage of women, relative to the null model, would go on to collaborate with a high percentage of women early-career researchers in their established period, compared with 47.2% of those who did not collaborate with a high percentage of women in their early career (Fig. 3a). For women, 81.8% of those whose early-career junior co-authors were composed of a high percentage of women would go on to collaborate with a high percentage of women in their established period, compared with 71.2% of women who did not collaborate with a high percentage of women in their early career. Furthermore, as with the mentorship data, this

substantial effect size is relatively stable over time for all gendered groups of researchers (Fig. 3b).

Paralleling our results from the mentorship data, we find that the gender diversity-association effect is greater for researchers with larger early-career collaboration networks (a proxy for group size; Fig. 3c). For researchers with 10 or more early-career co-authors when they themselves were early career, 69.0% of men who co-authored with a high percentage of women in their early career would go on to co-author with a high percentage of women in their established period, in contrast to 49.5% of those who did not co-author with a high percentage of women in the early-career period. For women, 84.0% of those who co-authored with a high percentage of women in their early career would go on to co-author with a high percentage of women in their established period, compared with 71.8% of those who did not co-author with a high percentage of women in their early career.

The gender diversity-association effects persist but show considerable variations across 138 countries worldwide (Fig. 3d,e). Among the top-20 countries by the number of established researchers, we find that all show significant gender diversity-association effects for both men and women researchers. We define $\Theta_c^{early}$ to indicate the early-career period and $\Theta_c^{est}$ to denote the established period for researchers from country $c$. We let $\Theta = 1$ represent the set of in-country researchers with a high percentage of women co-authors relative to the null model and $\Theta = 0$ denote the set of in-country researchers without a percentage of women co-authors. Thus, $P(\Theta_c^{est} = 1 | \Theta_c^{early} = 1)$ denotes the probability that researchers who co-authored with a high percentage of women in the early-career period go on to co-author

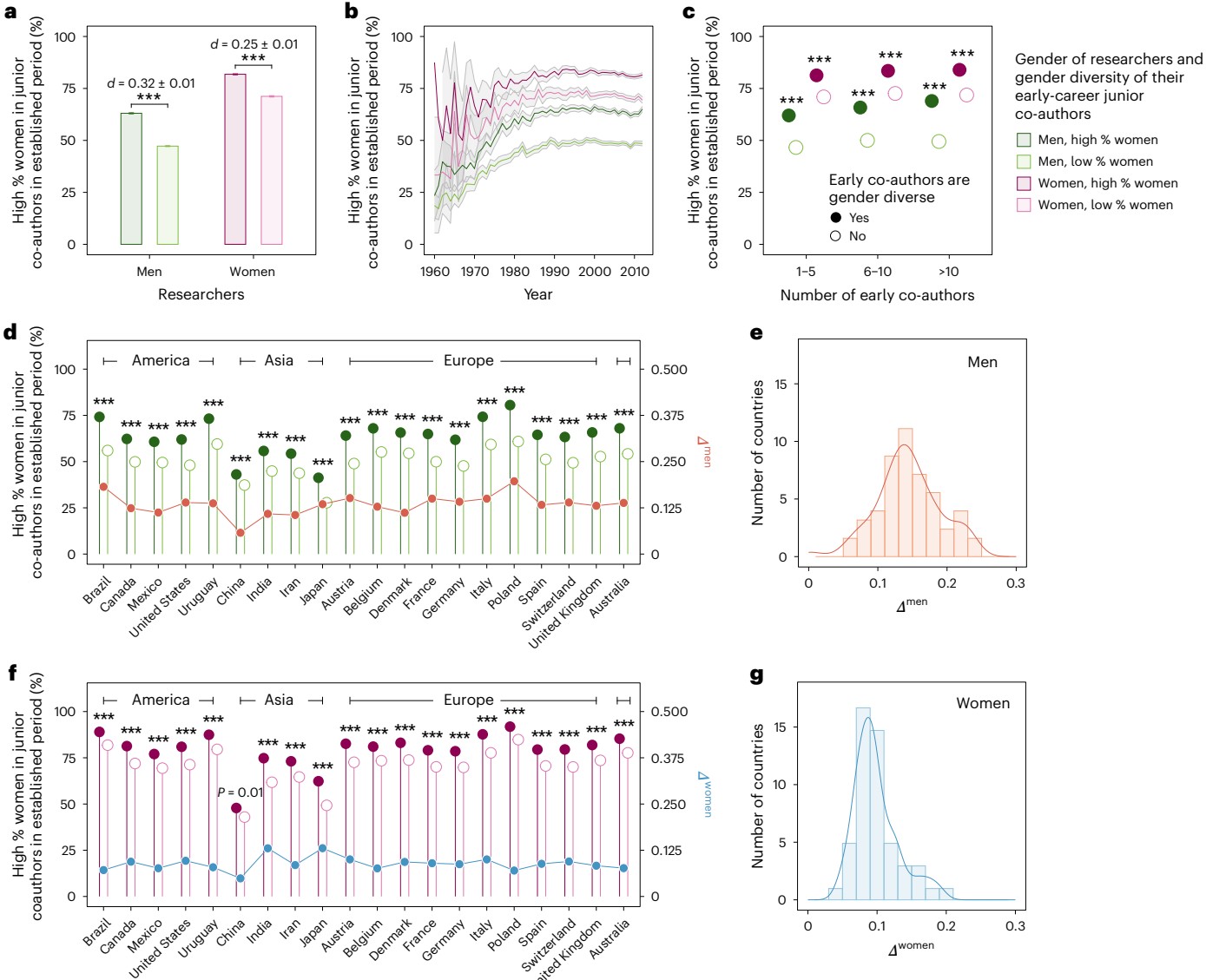

**Fig. 3 | Gender diversity association in the co-authorship networks of researchers.** Results of gender diversity-association analysis in the co-authorship networks relative to the randomized null model, showing how gender diversity of junior co-authors in the established period correlates with that of their early co-authors for individual researchers. The cohort of junior co-authors for a researcher is defined to have a high proportion of women if the percentage of women is greater than the baseline rate predicted by the null model. **a**, The interplay between the gender diversity of junior co-authors in the early-career and established periods of researchers, for men and women. **b**, Gender diversity-association effects in co-authorship networks over time, from 1960 to 2012, according to the first career years of individual researchers. **c**, Gender diversity-association effects by the number of junior co-authors in the early-career period. **d**,**f**, Gender diversity association across countries, for men (**d**) and women (**f**). **e**,**g**, Statistical distribution of gender diversity-association effect $\Delta_c$ of countries with at least 100 researchers, for men (**e**) and women (**g**). We select the top-20 countries with the largest number of established researchers and arrange them according to the continents they belong to. Bars in **a** and **c** represent mean values and error bars indicate 95% confidence intervals; the error bars in **c** are too small to be graphically visible. Lines denote mean values and shaded areas represent 95% confidence intervals in **b**. We report effect sizes using the Cohen's *d* statistic and use two-sided *t*-tests for comparisons. *P* values reported in the plot are adjusted by the Benjamini–Hochberg procedure for multiple comparisons. \*\*\**P* < 0.001.

with a high percentage of women in their established period. We then define the effect size for men researchers in country *c* as

$$\Delta_c^{\text{men}} \equiv P^{\text{men}}\left(\Theta_c^{\text{est}} = 1 | \Theta_c^{\text{early}} = 1\right) - P^{\text{men}}\left(\Theta_c^{\text{est}} = 1 | \Theta_c^{\text{early}} = 0\right). \quad (2)$$

where $P^{\text{men}}$ and $P^{\text{women}}$ define the conditional probability of a specific collaboration pattern for men and women researchers, respectively, and $\Delta$ refers to the difference of the probability of having gender diverse co-authors in the established period between groups of researchers with gender diverse or not early career co-authors. For most countries, we find $\Delta^{\text{men}} \approx 0.12$ and $\Delta^{\text{women}} \approx 0.10$, with the effect size being

larger for men researchers among American and European nations. For men whose early collaborators comprise a high percentage of women, American and European researchers are more likely to collaborate with a high percentage of women junior co-authors when they become established researchers. The gender diversity-association effect is particularly strong in Poland, Brazil, Italy and Uruguay, in terms of both the gender diversity of co-authors and the diversity-association effect size.

We use logistic regression models to quantify how having a high percentage of women co-authors in the early career of researchers predicts the outcome variable, a binary coding of whether the gender diversity of junior co-authors is high relative to the null model when

**Table 1 | Regression models to predict the gender diversity-association effect in mentorship and co-authorship networks**

| Model | a1 | a2 | b1 | b2 |
|---|---|---|---|---|
| Network | Mentorship | Mentorship | Co-authorship | Co-authorship |
| Regression type | Linear | Logistic | Linear | Logistic |
| Diversity using null model | No | Yes | No | Yes |
| (Intercept) | −0.181*** | −3.835*** | −0.323*** | −6.082*** |
| | (0.031) | (0.226) | (0.003) | (0.031) |
| Institutional prestige | −0.037* | −0.225 | −0.005*** | 0.004 |
| | (0.018) | (0.121) | (0.001) | (0.014) |
| Advisor group size | 0.001 | 0.004* | 0.000*** | 0.005*** |
| (Number of early junior co-authors) | (0.000) | (0.002) | (0.000) | (0.001) |
| Researcher is woman | 0.083*** | 0.565*** | 0.091*** | 0.710*** |
| | (0.009) | (0.050) | (0.001) | (0.007) |
| Women % by subfield | 0.830*** | 5.002*** | 0.806*** | 6.826*** |
| | (0.024) | (0.171) | (0.003) | (0.032) |
| Women % by country | 0.637*** | 4.779*** | 0.754*** | 6.226*** |
| | (0.070) | (0.525) | (0.005) | (0.051) |
| Early gender diversity | 0.118*** | 0.314*** | 0.086*** | 0.402*** |
| | (0.012) | (0.046) | (0.001) | (0.006) |

Under linear regression (models a1 and b1), the dependent variable is the proportion of women among advisees/junior co-authors of individual researchers in the established period. Under logistic regression (models a2 and b2), the dependent variable is a binary coding of whether individual researchers in the established period have a high percentage of women advisees/junior co-authors compared with the null model. The key variable is the early gender diversity, which measures the proportion of women among advisor's group members/junior co-authors in the early training period. It uses the raw women percentage in models a1 and b1, while it is a binary variable relative to the null model in models a2 and b2. Two-sided $t$-tests are used for multiple comparisons. Robust standard errors are given in parentheses. ***$P < 0.001$; *$P < 0.05$.

they become established researchers. Control variables include institutional prestige, the number of junior co-authors in the early-career period and several other variables, including researcher gender, the proportion of women junior researchers in the subfield and the proportion of women junior researchers by country to control for gender-related effects. We find that having a high percentage of women among early-career co-authors has a significantly positive relation with the outcome variable. The results are consistently significant for both the mentorship and co-authorship data across all regression models (Table 1 and Supplementary Tables 2 and 4). In particular, the odds ratio of having a high percentage of women among early co-authors is 1.495, suggesting that researchers whose early co-authors have a high percentage of women are more likely to cultivate gender-diverse research environments when they become established researchers (Table 1, model b2). The large coefficients on the women (%) by subfield and country variables suggest that demographic trends drive the majority of the gender diversification effect, with early group gender diversity playing a substantial and significant secondary role in further increasing group diversity in the established period.

### Racial diversity association

We measure the racial diversity of a specific group of researchers by the Shannon entropy score $h$ of the distribution of racial labels among its members (Methods). As a descriptive analysis of racial diversity association, we define junior co-authors of researcher $i$ to be racially diverse if $h_i > \langle h \rangle$, for both the early-career and the established periods. For researchers with racially diverse junior co-authors in the early-career period ($h_i^{\text{early}} > \langle h^{\text{early}} \rangle$), 63.7% go on to have racially diverse junior co-authors in the established period ($h_i^{\text{est}} > \langle h^{\text{est}} \rangle$). In comparison, for researchers without racially diverse junior co-authors in the early-career period ($h_i^{\text{early}} \leq \langle h^{\text{early}} \rangle$), only 36.3% go on to have racially diverse junior co-authors in the established period.

As with our assessment of gender diversity, we assess racial diversity-association effects relative to a null model of random mixing within the population while controlling for demographic constraints, but where we replace the gender labels with racial labels. We then define a researcher $i$'s group of collaborators to be racially diverse if the observed diversity exceeds the expected diversity under the model:

$$h_i > \langle h_i^{\text{null}} \rangle, \tag{3}$$

where $h_i$ is the entropy score for the racial distribution of co-authors and $\langle h_i^{\text{null}} \rangle$ is the average entropy score expected under the null model.

Among selected established researchers, 216,900 researchers had collaborated with racially diverse junior co-authors in the early-career period under the null model, composing 25.4% of all these researchers. We find that 40.8% whose early-career junior co-authors were racially diverse, relative to the null model, would go on to collaborate with a racially diverse set of early-career researchers in their established period, compared with 18.2% of those who did not have a racially diverse group of junior co-authors in their early career. (Fig. 4a). For researchers with more than 10 early-career co-authors, the rates are 43.8% versus 11.0%, respectively.

Arranging researchers over time according to their first year of publication, we also find that the racial diversity-association effect has increased over time, increasing from around 30% in the 1950s to nearly 45% after 2000 for established researchers with racially diverse early co-authors (Fig. 4b). For researchers without racially diverse early co-authors, the share with racially diverse early groups as established researchers also grew from the 1950s to about 1990, but then reversed course to decrease substantially, falling from just under 25% in the late 1980s to less than 15% by the 2010s. As such, the racial diversity-association effect becomes more evident over time.

Racial diversity association also shows substantial variation across geographic boundaries (Fig. 4c). Among the top-20 countries with the largest number of established researchers, we find that all show significant racial diversity-association effects. To illustrate country-level racial diversity-association effect sizes, we define $\Phi^{\text{early}}$ to indicate the

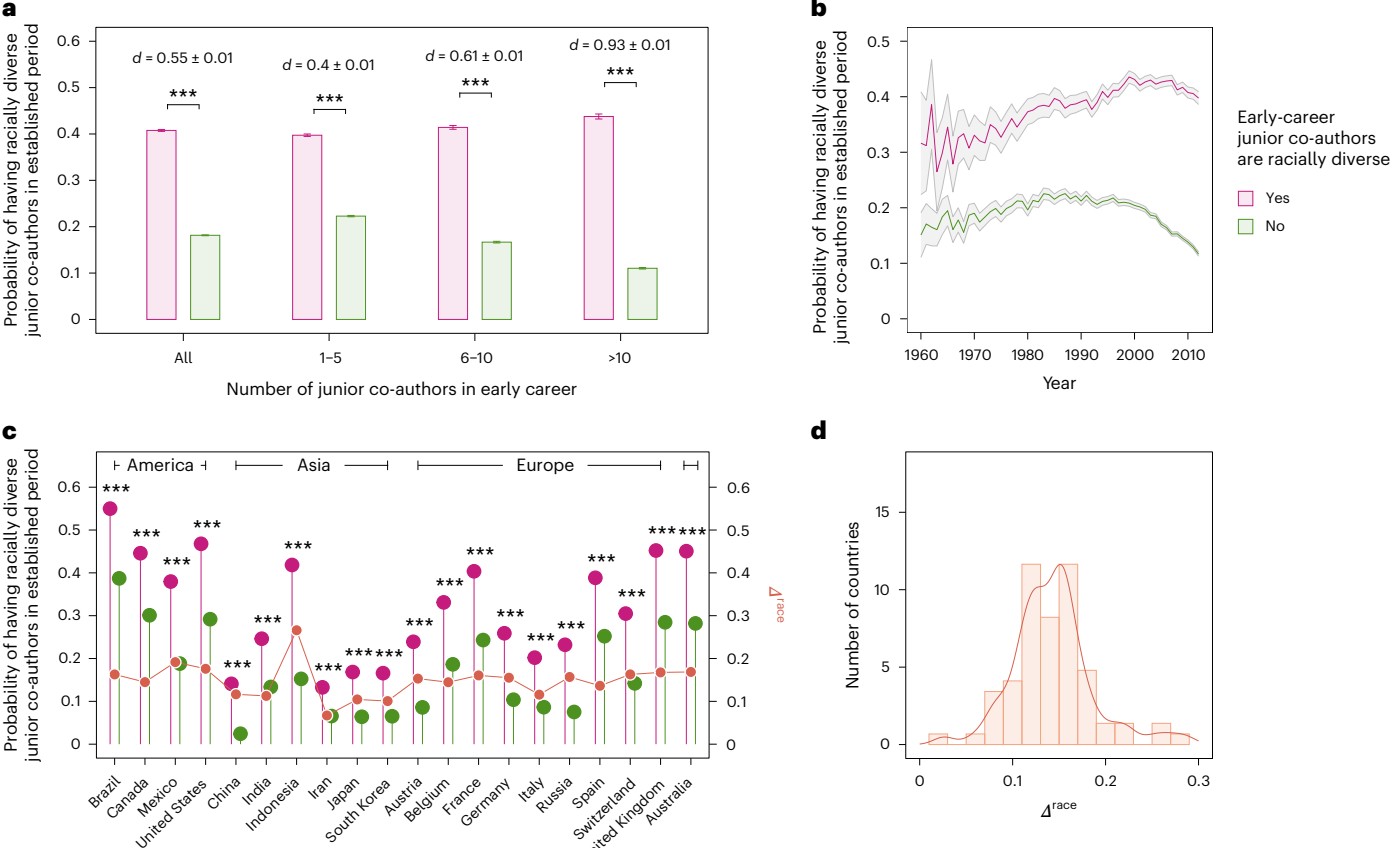

**Fig. 4 | Racial diversity-association effects in the co-authorship networks of researchers.** Racial diversity-association effects in co-authorship networks relative to the randomized null model, and how the racial diversity of junior co-authors in the established period is correlated with that of their early co-authors for individual researchers. The cohort of junior co-authors for a researcher is defined to be racially diverse if the racial entropy score is greater than the baseline rate predicted by the null model. **a**, The interplay between the racial diversity of junior co-authors in the early-career and established periods of researchers. **b**, Racial diversity-association effects in co-authorship networks over time, from 1960 to 2012, according to the first career years of individual

researchers. **c**, Racial diversity-association effects across countries, where we select the top-20 countries with the most researchers and arrange them according to the continents they belong to. **d**, Statistical distribution of racial diversity-association effect $\Delta_c$ of countries with at least 100 researchers. Bars in **a** represent mean values and error bars indicate 95% confidence intervals. Lines denote mean values and shaded areas represent 95% confidence intervals in **b**. We report effect sizes using the Cohen's $d$ statistic and use two-sided $t$-tests for comparisons. $P$ values reported in the plot are adjusted by the Benjamini–Hochberg procedure for multiple comparisons. ${}^{***}P < 0.001$.

early-career period and $\Phi^{\mathrm{est}}$ to denote the established period of an in-country researcher. We let $\Phi = 1$ to represent that the co-authors are racially diverse relative to the null model and $\Phi = 0$ to denote that the co-authors are not racially diverse. Thus, $P(\Phi_c^{\mathrm{est}} = 1 | \Phi_c^{\mathrm{early}} = 1)$ represents the probability that researchers with racially diverse junior co-authors in the early-career period go on to have racially diverse junior co-authors in the established period. Then we define the effect size as

$$\Delta_c^{\mathrm{race}} \equiv P\left(\Phi_c^{\mathrm{est}} = 1 | \Phi_c^{\mathrm{early}} = 1\right) - P\left(\Phi_c^{\mathrm{est}} = 1 | \Phi_c^{\mathrm{early}} = 0\right). \quad (4)$$

For the majority of selected countries, we estimate a range of $\Delta^{\mathrm{race}} \approx 0.1$– $0.2$. Among them, Indonesia has the largest, with $\Delta^{\mathrm{race}} \approx 0.3$, followed by North and South American countries such as Mexico, the United States and Brazil, and European countries such as the United Kingdom, France and Switzerland, all of which have highly diverse demographic populations. Most East Asian nations that have racially homogeneous populations have relatively smaller but still significant diversity-association effects. Hence, as we would expect, we find that scholars from countries with racially homogeneous populations are less likely to construct racially diverse collaboration networks in the early-career period

(Supplementary Fig. 23), and they are comparatively less likely to collaborate with racially diverse junior co-authors in the established period than those from racially diverse nations. Nevertheless, within a specific country, the racial diversity-association effect is consistently significant.

We use logistic regression models to examine how having racially diverse co-authors in the early career of researchers predicts the outcome variable, a binary coding of whether the racial diversity of junior co-authors is high relative to the null model when they become established researchers. Analogous to the gender analyses, we include several control variables and two other variables of the racial diversity of junior researchers by subfield and country to control for race-related effects. We find that having a high racial diversity among early-career co-authors has a significantly positive relation with the outcome variable. The odds ratio of having racially diverse junior co-authors in the early-career period is 1.917, suggesting that researchers who worked with racially diverse early-career junior co-authors are more likely to collaborate with a cohort of racially diverse junior researchers when they become established researchers (Table 2, model c2). The results are consistently significant across all regression models (Supplementary Table 6).

**Table 2 | Regression models to predict the racial diversity-association effect in the co-authorship networks**

| Model | c1 | c2 |
|---|---|---|
| (Intercept) | −0.127*** | −5.908*** |
|  | (0.003) | (0.040) |
| Institutional prestige | 0.032*** | 0.131*** |
|  | (0.001) | (0.014) |
| Number of early junior co-authors | −0.001*** | -0.014*** |
|  | (0.000) | (0.000) |
| Racial diversity by field | 0.181*** | 0.482*** |
|  | (0.005) | (0.052) |
| Racial diversity by country | 0.880*** | 5.611*** |
|  | (0.002) | (0.023) |
| High racial diversity in early co-authors | 0.123*** | 0.651*** |
|  | (0.001) | (0.006) |

Under linear regression (model c1), the dependent variable is the crude racial diversity among junior co-authors of individual researchers in the established period. Under logistic regression (model c2), the dependent variable is a binary coding of whether individual researchers in the established period have a high racial diversity among junior co-authors compared with the null model. The key variable is the racial diversity in early co-authors, which uses the raw racial diversity in model c1 and a binary coding relative to the null model in model c2. Two-sided $t$-tests are used for multiple comparisons. Robust standard errors are given in parentheses. ***$P < 0.001$.

## Discussion

The widespread diversity-association effects we observe could be driven by a variety of social processes. An important line of future work will be to identify the degree to which different mechanisms drive the observed empirical effects and to what extent the effects we observe are driven by selection. Regardless of the particular mechanism, diversity-association effects, over time, can operate like an amplifier, driving a field's overall representational diversity via a social reproduction process that increases the number of individual research groups that are representationally diverse. Diversity-association effects expand our inventory of the reasons that diversity trends are slow. Our results suggest that academic socialization encompasses social preferences in addition to scientific practices. Enhancing norms to diversify research groups and provide more opportunities for under-represented demographic groups serves to improve equity in academia and to direct great human capital into scientific innovation. It can also have long-term benefits in facilitating diversity trends, as junior researchers who had diverse collaboration experiences are more likely to build diverse research groups when they become established researchers, which contrasts with simple expectations of homophily.

Although our null model approach provides reasonable controls for a number of confounding factors, our study does not strongly identify causality or identify any particular mechanism beyond the association. Further investigations are needed to identify the particular sociological and psychological processes that underpin diversity-association effects among junior scholars and how they affect researchers' propensity to construct diverse environments when they become established researchers. Future studies using experimental designs of social group evolutions or large-scale empirical data of mentorship would help researchers understand dynamic patterns of diversity association and develop hypotheses based on social theories.

Another limitation stems from our use of name-based methods for gender and race assignment, which have known limitations in their accuracy, especially for minority women[48]. For instance, the US Census name data used in this study provide a reasonable representation of the general US population but are unlikely to capture the full diversity of name–demographic associations present in the global population of researchers. Even among US-affiliated researchers, the racial distribution from the US Census name data may not precisely reflect the racial distribution in the scientific workforce. Name-based gender assignment is relatively less accurate for Asian names, which probably reduces the magnitude of estimated diversity-association effects for Asian researchers[49,50]. Such discrepancies are likely to reduce the estimations of diversity obtainable in our analysis, suggesting that our measured effects may, on average, underestimate the true effect size. At the same time, all of our results should be interpreted within the context and specific accuracy of the demographic or geographic background of focal researchers.

To account for some of these limitations, especially researchers from under-represented racial groups, our name-based method was used in aggregate measures rather than individual classifications in the racial diversity analysis. This approximation restricts the ability to extend our model to studies of particular pairs of racial interactions. For instance, how does early-career racial diversity affect a White established researcher's collaboration with or mentorship of Black or Latine junior researchers? Do early-career interactions with minority students from one specific racial group influence how an established researcher interacts with junior researchers from other minority groups? Collecting large-scale self-identified race and ethnicity information to address these and related research questions is an important direction for future research.

Post-functionalist socialization mechanisms can shed light on other social diversity processes in science. There are many potential research questions regarding other aspects of scientific careers that may be influenced by early-career experiences with representational diversity. For example, do these experiences change what ideas scientists choose to study? And are there spillover effects to collaborators' groups? Do early-career experiences influence other scientific activities?

Several of the control factors in the regression analyses warrant future investigation to understand the underlying social mechanisms by which they influence diversity socialization (Tables 1 and 2). For the co-authorship network, both the gender and race analyses suggest that prestigious institutions may be better positioned or able to foster diversity socialization. More research is needed to understand how elite institutions diversify their workforce when they hire faculty and accept graduate students. In all the analyses, country of origin has a strong positive effect on diversity socialization, and more research is needed to investigate how the cultural and geographic boundaries that determine the demographic compositions and academic norms shape the gender and racial diversity socialization process in science.

More research is also needed to characterize other exogenous factors that may influence diversity-association effects in science. For example, how do the researchers' diversity preferences vary across institutions with different prestige levels? To what degree is diversity socialization related to culture, economic development or scientific infrastructure disparities across countries? Why does diversity socialization vary across fields? In addition to future quantitative work, careful qualitative work is needed to identify the specific social processes and choices that underpin the diversity associations we quantify.

Diversity-association effects that shape the demographic diversity of the scientific workforce also have policy implications. Government, funding agencies, universities and research institutions could allocate targeted fellowships to increase the demographic diversity of early-career collaborations, especially for those being trained in medium and large research groups. This may help attract junior researchers from diverse demographic backgrounds, and may have profound and long-lasting effects in promoting representational diversity in the scientific workforce.

## Methods

### Mentorship data

In this study, we use the mentorship dataset provided by ref. 46, in which mentor relationships were extracted from the Academic Family Tree project and researcher profiles were matched to the MAG data retrieved in September 2020. We refine and extract 339,744 advisor–advisee pairs that we could match their author and affiliation IDs in the MAG database and assign gender information using the name-based method. We retain 17,917 researchers whose advisors mentored at least 2 advisees during their training period and who mentored at least 1 advisee, and to whom we could assign a binary gender according to our gender classification method. Selecting research groups with at least two advisees guarantees that within-group collaborations and interactions occur among trainees during the training period.

There are several causes for the down-sample of retained researchers in the mentorship analysis. In the original dataset, just a fraction of researchers are selected after conforming to MAG identifications. The majority of researchers included in advisor–advisee relationships are active in recent years, many of whom appeared as trainees in the dataset but do not report whether they trained any of their own advisees later in the career. Moreover, advisor–advisee pairs are excluded from the analysis if they contain neither the start year nor the end year of the relationship. For advisor–advisee pairs without either the start year or the end year, we extrapolate missing data by assuming that each training period lasts for 5 years, for example, if the start year is 1992 and the end year is missing, we set the end year to 1997. To sum up, the selected 17,917 researchers formed 123,301 advisor–advisee relationships during their training period and 82,699 advisor–advisee relationships when they become established researchers.

### Co-authorship data

We use the publication data in the period of 1950–2021 from MAG, which was downloaded from the OpenAlex database. We include journal publications for 19 major disciplines including STEM, social science and humanities, and conference proceedings for computer science. Only papers that have affiliation information are considered. As in extremely large teams, the team assembly mechanisms may be different and co-authors are less likely to engage in effective social interactions, we consider papers that have at most 20 authors. We retain 30.6 million research papers after these data-filtering procedures and define co-authors as researchers who worked on the same papers. In our analysis, early-career researchers are within the first 3 years of their career, while established researchers have at least 6 years of publishing career by the time of collaboration, have at least 10 total career years and published at least 10 papers. In the gender analysis, we retain 562,494 such productive researchers who have collaborations with at least 1 junior co-author in both the early-career and the established periods. We also assign racial information to 855,526 productive researchers who have worked with at least 1 junior co-author in both the early-career and the established periods.

### Gender

We assign the gender labels to researchers based on the historical records of newborn baby names in the United States from the US Social Security Administration data[51]. Thus, our findings about gender diversity association are mostly applicable to researchers in North America or English-speaking countries. We only retain first names that have at least 95% confidence for a specific gender for the matching. Genders are assigned to author names in a binary way, as other genders are only available through self-identifications, which is an acknowledged limitation of this study. Compared with a more recent name-based gender classification method, our approach predicts the gender of 92% of researchers with about 99% accuracy[50]. Thus, the reliability of our gender classification results is comparable to other state-of-the-art methods. On the basis of these gender-name classifiers, we match 17,917

advisors in the mentorship data, among which 4,070 (22.7%) are women. In the co-authorship data, we also assign gender labels to 562,494 productive established researchers, of whom 181,467 (32.3%) are women.

### Race

The US Census data provide the number of people in each racial group for 162,253 most common family names. The racial categories used in the census data include White, Black, Asian and Pacific Islander, Asian, Latine, and people with two or more races. Given that people with two or more races account for only a minimal proportion of researchers[4], they are removed from the analysis. Instead of assigning names to the most probable racial group, we adopt a mixed model approach that assigns race distributions to researchers according to the associated probability of their family names to each racial group in the census data. Each researcher then contributes to the group diversity through the racial group distribution associated with its family name. Specifically, for a given author $i$, the probability of its co-authors belonging to racial group $j$ is defined as

$$p_i^j = \sum_{k=1}^{n_i} p_{i,k}^j / n_i, \tag{5}$$

where $p_{i,k}^j$ refers to the probability that $i$'s co-author $k$ belongs to racial group $j$ and $n_i$ is the number of $i$'s co-authors. Racial diversity is then defined as the Shannon entropy score

$$h_i = -\sum_{j=1}^{m} p_i^j \log\left(p_i^j\right), \tag{6}$$

where $m = 5$ is the number of racial groups used in the study.

### Null model

We use randomized network null models to estimate the expected gender and racial diversity. For the mentorship data, we first construct the original mentor–mentee network, and then reshuffle mentees controlling for a range of factors that may influence demographic composition of researchers, including time, subfield, institutional prestige and country. For the co-authorship data, we first construct the original co-authorship network using publication data. For each replication, we reshuffle the author order controlling for the aforementioned factors. After simulating 100 replications, we compute the average gender and racial diversity of individual researchers. We use the null model only to classify researchers into the high and low categories based on the sociodemographic diversity of their co-authors. For instance, in the racial analysis, we compare researcher $i$'s empirical racial diversity $h_i$ with the average diversity $\langle h_i^{\text{null}} \rangle$ in the null model replications to obtain a dummy variable of high and low diversity of its co-authors for both the early-career and established periods. For a specific country, we assemble all of its researchers and separate them into two groups according to the diversity of their early co-authors. Then we conduct a $t$-test for the co-author diversity in established period for these two groups of researchers, from which we obtain raw $P$ values. We then adjust these $P$ values using the Benjamini–Hochberg procedure for multiple comparisons by setting the false discovery rate to 0.05.

### Institutional prestige

The prestige of research institutions is defined as the $Z$ score of their highly cited publications, which refer to papers receiving the upper 5th percentile of citations 2 years after publication for a given year and subfield. For institution $i$, its prestige score in a given subfield (MAG level 1) is defined

$$p_i^{\text{inst}} = \frac{1}{100} \frac{N_i^{\text{high}} - \langle N^{\text{high}} \rangle}{\sigma / \sqrt{n^{\text{inst}}}}, \tag{7}$$

where $N_i^{high}$ is the number of highly cited papers produced by institution $i$, $\langle N^{high} \rangle$ is the average number of highly cited papers by all institutions, $\sigma$ is the standard deviation of highly cited papers and $n^{inst}$ is the number of institutions in the subfield. The coefficient 1/100 is to ensure that institutional prestige has a moderate coefficient value in the regression models. The institutional prestige score is subfield dependent but does not vary over time.

## Statistics and reproducibility

No statistical method was used to predetermine sample size. No data were excluded from the analyses.

## Data availability

The Microsoft Academic Graph data were obtained by following the guidelines at https://docs.microsoft.com/en-us/academic-services/graph/get-started-setup-provisioning. The mentorship dataset was obtained from ref. 46. The gender and family name data were obtained from the US Census Bureau website on 30 September 2022, at https://www.census.gov/topics/population/genealogy/data/2010_surnames.html. Source data are available with this paper.

## Code availability

Code is made available via Code Ocean at https://doi.org/10.24433/CO.5919525.v2 (ref. 52).

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

## Acknowledgements

This research was supported by the High Performance Computing Platform of Beihang University. W.L. was supported in part by National Key R&D Program of China (2022ZD0116800) and Program of National Natural Science Foundation of China (12441101, 62141605, 12201026, 11922102, 11871004). H.Z. was supported by Beijing Natural Science Foundation (Z230001). J.E.B. benefited from facilities and resources provided by the California Center for Population Research at UCLA (CCPR), which receives core support (P2C-HD041022) from the Eunice Kennedy Shriver National Institute of Child Health and Human Development (NICHD). A.C. was supported in part by Air Force Office of Scientific Research Award FA9550-19-1-0329.

## Author contributions

W.L., H.Z., J.E.B. and A.C. designed research and performed research. W.L. analyzed the data and made the visualizations. W.L., J.E.B. and A.C. wrote the paper.

## Competing interests

The authors declare no competing interests.

## Additional information

**Correspondence and requests for materials** should be addressed to Hongwei Zheng or Aaron Clauset.

