## [Peer Review File · Nature Computational Science]

Gender and racial diversity socialization in science

Corresponding Author: Professor Hongwei Zheng

Version 0:

Decision Letter:

**** Please ensure you delete the link to your author homepage in this e-mail if you wish to forward it to your co-authors. ****

Dear Professor Zheng,

Your manuscript "Gender and racial diversity socialization in science" has now been seen by 3 referees, whose comments are appended below. You will see that while they find your work of interest, they have raised points that need to be addressed before we can make a decision on publication.

The referees' reports seem to be quite clear. Naturally, we will need you to address ***all*** of the points raised.

While we ask you to address all of the points raised, the following points need to be substantially worked on:

- Please consider adding the missing related work pointed out by Referee #1.
- Please make sure to discuss the problem of promotion, as mentioned by Referee #1.
- Please make sure to provide enough details on the data and to add any missing controls in the null model, as mentioned by Referee #2.
- Please add a README file to the code so that reproduction and reusability instructions are clear.
- Note that the Article format allows up to 6 display items (figures and tables) in the main text. You currently have 4 display items, so if necessary, you could consider moving up to 2 figures/tables that you consider very relevant from SI to main text.

Please use the following link to submit your revised manuscript and a point-by-point response to the referees' comments (which should be in a separate document to any cover letter):

Link Redacted

**** This url links to your confidential homepage and associated information about manuscripts you may have submitted or be reviewing for us. If you wish to forward this e-mail to co-authors, please delete this link to your homepage first. ****

To aid in the review process, we would appreciate it if you could also provide a copy of your manuscript files that indicates your revisions by making use of Track Changes or similar mark-up tools. Please also ensure that all correspondence is marked with your Nature Computational Science reference number in the subject line.

In addition, please make sure to upload a Word Document or LaTeX version of your text, to assist us in the editorial stage.

Please make sure to update the Code Ocean capsule accordingly. If you have any issues when updating your Code Ocean capsule during the revision process, please email the Code Ocean support team Cc'ing me.

To improve transparency in authorship, we request that all authors identified as 'corresponding author' on published papers create and link their Open Researcher and Contributor Identifier (ORCID) with their account on the Manuscript Tracking System (MTS), prior to acceptance. ORCID helps the scientific community achieve unambiguous attribution of all scholarly contributions. You can create and link your ORCID from the home page of the MTS by clicking on 'Modify my Springer Nature account'. For more information please visit www.springernature.com/orcid.

We hope to receive your revised paper within three weeks. If you cannot send it within this time, please let us know.

Best,
Fernando

--

Fernando Chirigati, PhD
Chief Editor, Nature Computational Science
Nature Portfolio

Reviewers comments:

Reviewer #1 (Remarks to the Author):

This is an interesting paper that makes new contributions to knowledge about how race and gender are enmeshed in collaboration and mentorship dynamics.

At the front end it may be useful to be more explicit that there is also a problem of promotion; that even as women/faculty from underserved communities are hired, their retention rates are significantly lower. Since collaborative networks are key to publishing, the findings in this paper make important contributions to understanding not just representation but also retention.

The concept of "diversity association" as part of academic socialization is quite interesting, and compelling. One concern is that the authors are including a wide array of fields, including social sciences and humanities, that do not all work from the same "lab group" model. For example, a historian might have several students who never interact with one another. An appendix that explores differences between major fields might be useful.

These cites might be useful.

Brunsma, David L., David G. Embrick, and Jean H. Shin. 2017. "Graduate Students of Color: Race, Racism, and Mentoring in the White Waters of Academia." *Sociology of Race and Ethnicity* 3 (1): 1–13. doi: 10.1177/2332649216681565.
Maher, Michelle A., Joanna A. Gilmore, David F. Feldon, and Telesia E. Davis. 2013. "Cognitive Apprenticeship and the Supervision of Science and Engineering Research Assistants." *Journal of Research Practice* 9 (2): Article M5.

The methods and data are excellent. While there are considerable limits to the data, in particular the imputation of race and gender via names, the authors are careful not to overreach, and recognize the limits of what they can say. Generally, these issues create a conservative bias in the paper, as the authors note; the fuzziness of these measures should make it *more difficult* to find significant findings, yet the findings are significant, and they are likely underestimating these effects. The null model approach is very appealing, but could perhaps use another sentence or two to help the reader understand how, for example, it might look for a particular case.

The analysis is convincing. The findings, both around diversity of advisors' groups and own groups, and around collaboration networks, appear quite robust. As the authors note, this is an association, and they cannot make causal claims, but given the time order, it does seem that there is diversity socialization occurring that can explain the different levels of diversity in research groups and collaboration networks. It's unclear to me why Figs 3 and 4 appear at the end of the paper and Figures 1 and 2 embedded.

There are some pieces to the analysis that appear in the supplemental analyses that I wish were discussed a little more in text. It is not clear why, in the regressions predicting whether research has high% women advisees compared to null model, the dummy for post-2000 is significant and negative. The description of the prestige measure early in the paper suggested that prestige might matter. There is a sig positive effect in the model that predicts whether researchers have a high% women junior coauthors compared to the null model, as well as the model that predicts that researchers have racially diverse junior collaborators compared to the null, but not sure what that tells us - I would have expected the opposite. The country level data (Supp Fig 7) is extremely interesting, and could potentially use some more explication, but this might be better as a separate paper.

The subfield data (Supp Fig 11, 17) would be better presented in the text, since these dynamics are clearly different by fields where faculty have lab groups and those that do not. The difference in the diversity association by race versus gender as shown in Figs 17 and 11 were curious. While the race data is arguably less good than the gender, race appears to have a more powerful effect, which deserves further discussion.

As the researchers call for future research, they might suggest future qualitative work that works through the processes that underpin diversity association effects.

Generally, a strong paper, hope my comments are helpful.

Reviewer #2 (Remarks to the Author):

The manuscript, "Gender and Racial Diversity Socialization in Science," investigates how early-career experiences with gender and racial diversity in mentor-mentee networks and collaboration networks influence the diversity of researchers' collaboration networks as they advance in their careers. It focuses on understanding the "diversity association effect," wherein early exposure to diverse environments may shape researchers' long-term preferences and behaviors regarding diversity. The study uses two datasets for analysis: (1) a mentor-mentee dataset covering approximately 17,900 researchers and (2) a co-authorship dataset encompassing over 560,000 scholars. A computational null model is proposed to support the authors' claims, controlling for confounding factors such as subfield, country, and institutional prestige.

The question I have about the manuscript are quite basic. Many of them reside in data and empirical analysis.

The data description:

Several aspects of the data remain unclear after reviewing the manuscript and Supplementary Information (SI). For instance, the authors did not describe the proportion of researchers without complete first names. Are those researchers with only first initials significant enough to potentially alter the main findings? Additionally, it would be helpful to include basic descriptive statistics in the SI, such as the gender distribution across the 19 fields and across different cohorts (e.g., based on the year of a scholar's first publication). These details would provide readers with better context for understanding the data.

I found the use of shapes and colors in Figure 1 to represent gender and the focal author somewhat confusing. The color pair assigned to men and women feels counterintuitive. Many other studies I have read use the opposite color combination. Revisiting the visualization design may enhance clarity.

The empirical analysis/mechanism:

First, there appear to be some missing controls in the null model. For instance, while the model considers pairs active in the same year, it might also need to account for cohort effects (i.e., the year an author published their first paper). Additionally, the cumulative citations of the senior author at the time of link formation could be a valuable factor to consider when rewiring links.

Secondly, some results in the regression analysis (e.g., SI Table 1 and Table 2) are unclear. For example, why do some variables have zero coefficients and zero standard errors? If these are fixed-effect variables, their coefficients should not be reported in the table. Clarifying this issue would help avoid confusion.

Thirdly, there are a few typos in the SI. For instance, in the section describing the null model for co-authorship networks, page 19 states: "We repeat this process for all advisor-advisee pairs for each iteration of the network null model randomization." This appears to be a direct copy-paste from SI section 1.2 and should be revised. Please ensure the manuscript and SI are carefully copy-edited.

Fourth, while the paper effectively demonstrates how early-career experiences with gender and racial diversity influence long-term collaboration behaviors, it could also address a critical question: do these preferences benefit science? For instance, do such collaborations lead to high-impact and novel scientific discoveries? Investigating these outcomes could further strengthen the paper's contribution.

I hope these comments are helpful in the development of your manuscript.

Reviewer #3 (Remarks to the Author):

This manuscript uses a null model to explore the "diversity association" between a researcher's mentorship and collaboration during their training and establishment phases, with a focus on gender and racial diversity. The rewire method is well-established in the field of physics, and the results are statistically robust and convincing, and presented in a clear form.

I have one minor suggestion. While the authors have provided qualitative conclusion supported by strong evidence regarding the presence of a diversity association, I wonder if they could expand the analysis to quantify this effect. Specifically, could the authors explore the expected impact of a 1% increase in gender/racial diversity during the training phase on the diversity observed in the establishment phase? Is the response linear, or does it exhibit a saturation point at higher levels of diversity?

Additionally, there is a typo: I believe "racial" in Line 315 should be "gender."

Overall, this is a well-prepared and thoughtfully presented manuscript, and I look forward to seeing it published.

Reviewer #3 (Remarks on code availability):

The code is accessible. It would be helpful if the authors provide a readme file.

Version 1:

Decision Letter:

Our ref: NATCOMPUTSCI-24-2778A

3rd March 2025

Dear Dr. Zheng,

Thank you for submitting your revised manuscript "Gender and racial diversity socialization in science" (NATCOMPUTSCI-24-2778A). It has now been seen by the original referees and their comments are below. The reviewers find that the paper has improved in revision, and therefore we'll be happy in principle to publish it in Nature Computational Science, pending minor revisions to satisfy the referees' final requests and to comply with our editorial and formatting guidelines.

TRANSPARENT PEER REVIEW

Nature Computational Science offers a transparent peer review option for original research manuscripts. We encourage increased transparency in peer review by publishing the reviewer comments, author rebuttal letters and editorial decision letters if the authors agree. Such peer review material is made available as a supplementary peer review file. **Please remember to choose, using the manuscript system, whether or not you want to participate in transparent peer review.**

Thank you again for your interest in Nature Computational Science. Please do not hesitate to contact me if you have any questions.

Sincerely,
Fernando

--

Fernando Chirigati, PhD
Chief Editor, Nature Computational Science
Nature Portfolio

ORCID

Reviewer #1 (Remarks to the Author):

The authors have systematically revised the paper, addressing the comments of the reviewers thoughtfully. As a result, what was a strong paper, is stronger now.

I appreciate the careful attention to detail, and the important contributions the paper makes.

Reviewer #2 (Remarks to the Author):

Thank you for addressing my previous comments. I am satisfied with the current revised version. However, one minor issue may require your attention: the selection of thresholds for classifying author cumulative citations. Specifically, the rationale behind choosing 100, 300, and 1,000 as thresholds needs further clarification. Are these values based on the 25th, 50th, and 75th percentiles of the distribution? Providing a justification for these cutoffs would strengthen the methodological transparency of your analysis.

Reviewer #3 (Remarks to the Author):

The authors have cleared all my suggestions and concerns. I'll be happy to see this manuscript on Nature Computational Science.

Junming Huang

Version 2:

Decision Letter:

Dear Professor Zheng,

We are pleased to inform you that your Article "Gender and racial diversity socialization in science" has now been accepted for publication in Nature Computational Science.

Once your manuscript is typeset, you will receive an email with a link to choose the appropriate publishing options for your paper and our Author Services team will be in touch regarding any additional information that may be required.

Authors may need to take specific actions to achieve [compliance with funder and institutional open access mandates](https://www.springernature.com/gp/open-research/funding/policy-compliance-faqs). If your research is supported by a funder that requires immediate open access (e.g. according to [Plan S principles](https://www.springernature.com/gp/open-research/plan-s-compliance)) then you should select the gold OA route, and we will direct you to the compliant route where possible. For authors selecting the subscription publication route, the journal's standard licensing terms will need to be accepted, including [self-archiving policies](https://www.springernature.com/gp/open-research/policies/journal-policies). Those licensing terms will supersede any other terms that the author or any third party may assert apply to any version of the manuscript.

Acceptance of your manuscript is conditional on all authors' agreement with our publication policies (see <https://www.nature.com/natcomputsci/for-authors>). In particular your manuscript must not be published elsewhere and there must be no announcement of the work to any media outlet until the publication date (the day on which it is uploaded onto our web site).

Before your manuscript is typeset, we will edit the text to ensure it is intelligible to our wide readership and conforms to house style. We look particularly carefully at the titles of all papers to ensure that they are relatively brief and understandable.

Once your manuscript is typeset, you will receive a link to your electronic proof via email with a request to make any corrections within 48 hours. If, when you receive your proof, you cannot meet this deadline, please inform us at rjsproduction@springernature.com immediately.

If you have queries at any point during the production process then please contact the production team at rjsproduction@springernature.com.

We welcome the submission of potential cover material (including a short caption of around 40 words) related to your manuscript; suggestions should be sent to Nature Computational Science as electronic files (the image should be 300 dpi at 210 x 297 mm in either TIFF or JPEG format). We also welcome suggestions for the Hero Image, which appears at the top of our [home page](http://www.nature.com/natcomputsci); these should be 72 dpi at 1400 x 400 pixels in JPEG format. Please note that such pictures should be selected more for their aesthetic appeal than for their scientific content, and that colour images work better than black and white or grayscale images. Please do not try to design a cover with the Nature Computational Science logo etc., and please do not submit composites of images related to your work. I am sure you will understand that we cannot make any promise as to whether any of your suggestions might be selected for the cover of the journal.

Best regards,
Fernando

--

Fernando Chirigati, PhD
Chief Editor, Nature Computational Science
Nature Portfolio

P.S. Click on the following link if you would like to recommend Nature Computational Science to your librarian: https://www.springernature.com/gp/librarians/recommend-to-your-library

** Visit the Springer Nature Editorial and Publishing website at www.springernature.com/editorial-and-publishing-jobs for more information about our career opportunities. If you have any questions please click here. **

Response to the reviews of manuscript NATCOMPUTSCI-24-2778: “Gender and racial diversity socialization in science”

Dear Editors and Reviewers,

Thank you for your detailed attention to our manuscript, from technical concerns to broad ideas. Your comments have led us to revise the paper’s text and several figures for clarity and detail, include a richer body of literature, and insert additional statistical analyses and robustness tests. While the results of the paper have not qualitatively changed, the paper is now stronger. We hope that you find this revision significantly improved as a result of the changes, and will consider recommending it for publication.

In the document below, we have presented text in grey *italics* to quote a given review or editorial comment verbatim, in its entirety, and in the order of the review. Following each quote, we address the comment by discussing how we improved our manuscript to the expected satisfaction of the Reviewers and Editor. As the Editor and Reviewers will see, we took all comments generated in the review process serious, thoroughly addressed every one of them, and made revisions to our manuscript as a result.

Our responses are organized into the following sections:

- Response to the Editor
- Response to Reviewer 1
- Response to Reviewer 2
- Response to Reviewer 3

Sincerely,

Weihua Li, Hongwei Zheng, Jennie E. Brand & Aaron Clauset

Response to the Editor

Your manuscript "Gender and racial diversity socialization in science" has now been seen by 3 referees, whose comments are appended below. You will see that while they find your work of interest, they have raised points that need to be addressed before we can make a decision on publication.

*The referees' reports seem to be quite clear. Naturally, we will need you to address *all* of the points raised.*

Addressed: We thank the Editor for the opportunity to revise and resubmit our manuscript. We have carefully addressed every point raised by three Reviewers in the main paper and the Supplementary Information, and we report the details of our revision in this response memo. In this Section, we summarize and reference how we have addressed the following points highlighted by the Editor. We believe that we have successfully addressed all the points raised by the Editor and Reviewers, and have improved the quality of our manuscript substantially.

While we ask you to address all of the points raised, the following points need to be substantially worked on:

- Please consider adding the missing related work pointed out by Referee 1.

Addressed: We have added the missing related references recommended by Reviewer 1, and discussed them in the Introduction of the main paper: *"During doctoral training, academic socialization is a key social mechanism by which trainees learn the knowledge, values, and interpersonal skills that facilitate role performance and prepare them for entry into professional academic careers¹⁻⁵. Specifically, it encompasses the process of acquiring specific research skills, learning their field's norms and standards for scientific inquiry, developing a scholarly perspective, and learning to be an independent scholar⁶. Academic socialization is mediated by mentorship in the doctoral training period, the quality of which is a strong predictor of trainees' development of scientific skills, social integration, and mental health^{7,8}."* In another paragraph, we added: *"Academic socialization may be particularly relevant for underrepresented cohorts of students, as empirical studies have suggested that mentoring is often inadequate for graduate students of color⁷."*

- Please make sure to discuss the problem of promotion, as mentioned by Referee 1.

Addressed: We have further discussed the issue of promotion and retention in academia, which is a key component and outcome of the diversity socialization process. In the Introduction, we write about retention bias: *“Even as women or underrepresented faculty are hired, they may leave academia at higher rates, especially at lower prestige institutions^{9,10}.”* In another part of the Introduction, we also discuss the problem of retention and promotion for women faculty: *“Biases in faculty hiring can further decrease women’s representation^{10,11}, and even when they are hired, their retention rates are significantly lower, via work-life incongruencies like the unequal impact of parenthood^{12–15}, and workplace climates that favor men^{9,16}. Underrepresented minority faculty can also experience double standards in promotion and tenure decisions at US universities, in how their scholarly productivity and impact are judged by committees¹⁷.”*

- Please make sure to provide enough details on the data and to add any missing controls in the null model, as mentioned by Referee 2.

Addressed: We thank the Reviewer for this suggestion. We now conduct robustness tests with these two new control factors and we have added a new section in the Supplementary Information to provide more details on the data, especially regarding the author names and the name-based gender assignment. This method does not apply to authors that use only initials of first names, which we show in Fig. R3a. Among the researchers that we assign gender, we show the proportion of women by field in Fig. R3b. The proportion of women researchers has been continually increasing over the past five decades, which we show in Fig. R3c. These details give us more information about the author’s name and gender information, and statistical variations over time and across fields.

We have also added missing controls in the null model and incorporated the results of robustness tests in the Supplementary Information. For the gender diversity analysis in the coauthorship network, we conduct a robustness test by incorporating two additional control variables into the null model, which is the first publishing year of authors and the career cumulative citation counts of senior authors. In the new null models, we find that the gender diversity association effect persists for established researchers in both the aggregate analysis and the time dynamics, which we show in Fig. R7. We also find that the racial diversity association effect persists for established researchers in both the aggregate analysis and the time dynamics, and present the results in Fig. R8.

- Please add a README file to the code so that reproduction and reusability instructions are clear.

Addressed: We have prepared a README file in the Code Ocean console where we deposit our code, which we also attach to the end of this response memo (see APPENDIX I). We think the README file can help the reader understand the background and functionality of the null model, and the data and software necessary to reproduce and replicate the models.

- Note that the Article format allows up to 6 display items (figures and tables) in the main text. Your currently have 4 display items, so if necessary, you could consider moving up to 2 figures/tables that you consider very relevant from SI to main text.

Addressed: We thank the Editor for reminding us of the journal format requirements. We have added two regression tables to the main text of our manuscript (Tables 1 and 2), and the main paper now contains 6 displayable items, including 4 figures and 2 tables. For the gender diversity socialization analysis, we include the linear regression and the logistic regression models for both the mentorship and coauthorship networks using the full control variable set (Table R1). For the racial analysis, we include the linear regression and the logistic regression models for the coauthorship network using the full control variable set (Table R2). Including these regression analyses provides more convincing evidence of the diversity association effects in scientific networks.

Table R1: Regression models to predict the gender diversity association effect in the mentorship and coauthorship networks. For the linear regression models a1 and b1, the dependent variable is the proportion of women among advisees/junior coauthors of individual researchers in the established period. For the logistic regression models a2 and b2, the dependent variable is a binary coding of whether individual researchers in the established period have a high percentage of women advisees/junior coauthors compared to the null model. The key variable is early gender diversity, which measures the proportion of women among advisor's group members/junior coauthors in the early training period. It uses the raw women percentage in models a1 and b1, while it is a binary variable relative to the null model in models a2 and b2.

Model:	a1	a2	b1	b2
Network:	Mentorship	Mentorship	Coauthorship	Coauthorship
Regression type:	Linear	Logistic	Linear	Logistic
Diversity using null model:	No	Yes	No	Yes
(Intercept)	-0.181*** (0.031)	-3.835*** (0.226)	-0.323*** (0.003)	-6.082*** (0.031)
Institutional prestige	-0.037* (0.018)	-0.225 (0.121)	-0.005*** (0.001)	0.004 (0.014)
Advisor group size (No. early junior coauthors)	0.001 (0.000)	0.004* (0.002)	0.000*** (0.000)	0.005*** (0.001)
Researcher is woman	0.083*** (0.009)	0.565*** (0.050)	0.091*** (0.001)	0.710*** (0.007)
Women(%) by subfield	0.830*** (0.024)	5.002*** (0.171)	0.806*** (0.003)	6.826*** (0.032)
Women(%) by country	0.637*** (0.070)	4.779*** (0.525)	0.754*** (0.005)	6.226*** (0.051)
Early gender diversity	0.118*** (0.012)	0.314*** (0.046)	0.086*** (0.001)	0.402*** (0.006)

Robust standard-errors in parentheses

Signif. Codes: *** $p < 0.001$; ** $p < 0.01$; * $p < 0.05$

Table R2: **Regression models to predict the racial diversity association effect in the coauthorship networks.** For the linear regression model c1, the dependent variable is the crude racial diversity among junior coauthors of individual researchers in the established period. For the logistic regression model c2, the dependent variable is a binary coding of whether individual researchers in the established period have a high racial diversity among junior coauthors compared to the null model. The key variable is the racial diversity in early coauthors, which uses the raw racial diversity in model c1 and a binary coding relative to the null model in model c2.

Model:	c1	c2
Regression type:	Linear	Logistic
Diversity using null model:	No	Yes
(Intercept)	-0.127*** (0.003)	-5.908*** (0.040)
Institutional prestige	0.032*** (0.001)	0.131*** (0.014)
No. early junior coauthors	-0.001*** (0.000)	-0.014*** (0.000)
Racial diversity by field	0.181*** (0.005)	0.482*** (0.052)
Racial diversity by country	0.880*** (0.002)	5.611*** (0.023)
High racial diversity in early coauthors	0.123*** (0.001)	0.651*** (0.006)

Robust standard-errors in parentheses

Signif. Codes: *** $p < 0.001$; ** $p < 0.01$; * $p < 0.05$

Response to Reviewer 1

This is an interesting paper that makes new contributions to knowledge about how race and gender are enmeshed in collaboration and mentorship dynamics.

Addressed: Thank you for the positive endorsement of our work. We have carefully addressed all your suggestions, and made the changes in the revised manuscript and the Supplementary Information file. We feel that these efforts have substantially improved both the sociological theory and data analytics of our work.

At the front end it may be useful to be more explicit that there is also a problem of promotion; that even as women/faculty from underserved communities are hired, their retention rates are significantly lower. Since collaborative networks are key to publishing, the findings in this paper make important contributions to understanding not just representation but also retention.

Addressed: We agree with the Reviewer that there is also a problem of promotion, and our work may improve our understanding of both representation and retention. In the Introduction of the main paper, we further discuss promotion and retention in academia, which is a key component and outcome of the diversity socialization process. We write about retention bias: *“And, even as women or underrepresented faculty are hired, they often, but not always, leave academia at higher rates, especially at lower prestige institutions^{9,10}, and can face double-standards in promotion evaluations¹⁷.”* In another part of the Introduction, we also discuss the problem of retention and promotion of women faculty: *“Biases in faculty hiring can further decrease women’s representation^{10,11}, and even when they are hired, their retention rates are significantly lower, via work-life incongruencies like the unequal impact of parenthood^{12–15}, and workplace climates that favor men^{9,16}. Underrepresented minority faculty can also experience double standards in promotion and tenure decisions at US universities, in how their scholarly productivity and impact are judged by committees¹⁷.”*

The concept of “diversity association” as part of academic socialization is quite interesting, and compelling. One concern is that the authors are including a wide array of fields, including social sciences and humanities, that do not all work from the same “lab group” model. For

example, a historian might have several students who never interact with one another. An appendix that explores differences between major fields might be useful.

Addressed: The Reviewer has raised a very good point about how diversity associations may diverge across research fields. The comparison proposed by the Reviewer between a research group in the social sciences and humanities vs. STEM labs is important. The sociological theory we describe and our data analysis may be applied across fields, but will be most applicable to fields with collaborative norms among advisors and advisees, i.e., fields where advisors and advisees coauthor publications together¹⁸. This includes STEM fields with a standard “lab group” model, as well as some social sciences where research groups and collaboration norms are increasingly common, e.g., business, psychology, and quantitative social sciences. However, among researchers in fields where widespread collaboration norms are more limited, e.g., researchers more broadly in economics, political science, history, sociology, etc., there will be fewer coauthored publications available to our analyses. This suggests some degree of field heterogeneity, but also may limit the statistical power of our analyses for these researchers.

We add an additional analysis to the Supplementary Information to further explore field differences of gender diversity association effects. Specifically, we examine how the gender diversity association effect varies in four major domains, i.e., arts and humanities, engineering and mathematics, natural sciences, and social sciences (Supplementary Figure 15). The gender diversity association effect is not significant in arts and humanities, especially for researchers with a large number of early coauthors. In contrast, natural sciences have substantial diversity association effect for both men and women. This may be due to the group dynamics difference in these fields, where students tend to work on individual research projects in arts and humanities, while there is more collaborative work for students in natural sciences, especially in biology and physics, where large research groups have become more common.

We further examine how the racial diversity association effect varies in four major domains, defined the same as in the gender analysis (Fig. R2). The racial diversity association effect is not significant in arts and humanities, which includes arts, history, and philosophy, especially for researchers with 10 or more early coauthors. In other domains, including engineering and mathematics, natural sciences, and social sciences, the racial diversity association effect is fairly consistent, suggesting that research fields may have a moderate influence on racial diversity socialization. These results help us better understand how the academic practice and lab culture in different fields can shape the diversity association effects.

These cites might be useful.

Figure R1: **Gender diversity association effect in the coauthorship network by research domains and number of early coauthors.** We show how the gender diversity association effect varies by research fields in four domains, i.e., arts and humanities, engineering and mathematics, natural sciences, and social sciences. ($***p < 0.001$; $**p < 0.01$; $*p < 0.05$).

Brunsmas, David L., David G. Embrick, and Jean H. Shin. 2017. "Graduate Students of Color: Race, Racism, and Mentoring in the White Waters of Academia." *Sociology of Race and Ethnicity* 3 (1): 1–13. doi: 10.1177/2332649216681565.

Maher, Michelle A., Joanna A. Gilmore, David F. Feldon, and Telesia E. Davis. 2013. "Cognitive Apprenticeship and the Supervision of Science and Engineering Research Assistants." *Journal of Research Practice* 9 (2): Article M5.

Addressed: These references enrich the scope of our sociological theory and discussions of mentorship and supervision in the doctoral training of students. We have added the references recommended by the Reviewer, and discussed them in the Introduction of the main paper: "During doctoral training, academic socialization is a key social mechanism by which trainees learn the knowledge, values, and interpersonal skills that facilitate role performance and prepare them for entry into professional academic careers¹⁻⁵. Specifically, it encompasses the process of acquiring specific research skills, learning their field's norms and standards for scientific inquiry, developing a scholarly perspective, and learning to be an independent scholar⁶. Academic socialization is mediated by mentorship in the doctoral training period, the quality of which is a strong predictor of trainees' development of scientific skills, social

Figure R2: **Racial diversity association effect in the coauthorship network by research domains and the number of early coauthors.** We show how the racial diversity association effect varies by research fields in four domains, i.e., arts and humanities, engineering and mathematics, natural sciences, and social sciences. (***) $p < 0.001$; (***) $p < 0.01$; (*) $p < 0.05$).

integration, and mental health^{7,8}.” In another paragraph, we added: “Academic socialization may be particularly relevant for underrepresented cohorts of students, as research suggests that mentoring is often inadequate for graduate students of color⁷.”

*The methods and data are excellent. While there are considerable limits to the data, in particular the imputation of race and gender via names, the authors are careful not to overreach, and recognize the limits of what they can say. Generally, these issues create a conservative bias in the paper, as the authors note; the fuzziness of these measures should make it *more difficult* to find significant findings, yet the findings are significant, and they are likely underestimating these effects. The null model approach is very appealing, but could perhaps use another sentence or two to help the reader understand how, for example, it might look for a particular case.*

Addressed: Thank you for acknowledging our methods and analytics in this study. As the Reviewer correctly notes, there are limitations to the data, especially regarding the gender

and race imputation by author names. These limitations can create a conservative bias of the real diversity association effects. We also noted in the Discussion of the main paper. The data analytics reported in the figures and the regression tables provide strong evidence of the diversity association effects in science, for both the mentorship and coauthorship networks.

The Reviewer makes a very good suggestion here to look for a particular case to help explain the necessity of a null model. In the main paper, we wrote: “For example, a medical research group of 7 students with 3 women is common, as gender representation in the medical workforce is now close to parity. In contrast, a computer science research group with 6 members and 2 women may be considered gender diverse, because women’s representation in computer science has grown over the past several decades to be around 20-30%.”

In the Supplementary Information, we also include an example to explain the null model. As we discuss in the main text, the demographic composition of the scientific workforce can vary, and it is dependent on a number of exogenous factors such as time, subfield, country, and institutional prestige. For example, an engineering department with 10% women faculty in 1950 could be quite gender-diverse, while in 2020 most of these departments hire more than 15% of women among their faculty¹⁹. Field of study can be another critical factor in determining the level of expected gender diversity. A medical research group with 40% women is common, as medicine is a fairly gender-balanced field. In contrast, a research group in computer science with 30% women is quite gender diverse, as computer science is heavily dominated by men.

The analysis is convincing. The findings, both around diversity of advisors’ groups and own groups, and around collaboration networks, appear quite robust. As the authors note, this is an association, and they cannot make causal claims, but given the time order, it does seem that there is diversity socialization occurring that can explain the different levels of diversity in research groups and collaboration networks. It’s unclear to me why Figs 3 and 4 appear at the end of the paper and Figures 1 and 2 embedded.

Addressed: Thank you for pointing out this figure display issue in the main paper. We have fixed it in the new manuscript. This was an unforeseen error with Latex’s arrangement of figures. We tried a few ways to fix this and finally chose to use a smaller font size for the captions of all figures in the main text, which allows Latex to embed the figures in the proper positions of the article where they are mentioned and discussed.

There are some pieces to the analysis that appear in the supplemental analyses that I wish were discussed a little more in text. It is not clear why, in the regressions predicting whether research has high% women advisees compared to null model, the dummy for post-2000 is significant and negative. The description of the prestige measure early in the paper suggested that prestige might matter. There is a sig positive effect in the model that predicts whether researchers have a high% women junior coauthors compared to the null model, as well as the model that predicts that researchers have racially diverse junior collaborators compared to the null, but not sure what that tells us - I would have expected the opposite. The country level data (Supp Fig 7) is extremely interesting, and could potentially use some more explication, but this might be better as a separate paper.

Addressed: We thank the Reviewer for raising these questions. After careful consideration, we realized that the original specification of the regression model was problematic, as temporal variation was being integrated over in several of the control variables, which we believe led to the unexpected negative coefficient on the post-2000 variable. In the revised manuscript, we present new regression models in which we more carefully account for the effects of time within our analytic framework, so that control variables are defined either on a researcher's early or established career period, which naturally accounts for variations over time overall, such as demographic trends toward greater representation for women in a field or country.

We expected that institutional prestige may affect diversity socialization. On the one hand, researchers from elite institutions may be more likely to foster diverse environments as they may take the lead in advocating equity, diversity, and inclusion. On the other hand, elite institutions may have a greater incentive to recruit more privileged cohorts of researchers as the expectations of brilliance are overwhelming there²⁰. Elite institutions may have more advocates and implement more tangible measures to improve diversity and inclusion among their students and faculty. Examining which factors may dominate how elite institutions diversify their workforce when they hire faculty and accept graduate students could be a potential direction of future research.

Country of origin has a profound influence on both gender and racial diversity socializations, yet the pattern and magnitude of influence appear to be different. For the gender analysis, some Asian countries have relatively less gendered effects, but this may be due to the name-based gender assignment procedure we used to infer author gender. More research is needed to investigate the real gender diversity association effects in these countries with better author gender information data. Racial diversity is country-specific, and researchers from more racially diverse countries have larger racial diversity association effects. Investigating how the demographic composition and academic norms shape gender and racial diversity socialization in different cultures and countries might be an interesting direction for future work.

We summarize these points into a new paragraph in the Discussion: “ *Several of the control factors in the regression analyses warrant future investigation to understand the underlying social mechanisms by which they influence diversity socialization (Tables 1 and 2). For instance, because the composition of the scientific workforce has changed substantially over our study interval, we expected time to have a significant and positive effect on gender diversity socialization. Yet, time does not appear to have a consistent effect in these models, as it may be collinear with other factors. For the coauthorship network, both the gender and race analyses suggest that prestigious institutions may be better positioned or able to foster diversity socialization. More research is needed to understand how elite institutions diversify their workforce when they hire faculty and accept graduate students. In all the analyses, country of origin has a strong positive effect on diversity socialization, and more research is needed to investigate how the cultural and geographic boundaries that determine the demographic compositions and academic norms shape gender and racial diversity socialization process in science.*”

The subfield data (Supp Fig 11, 17) would be better presented in the text, since these dynamics are clearly different by fields where faculty have lab groups and those that do not. The difference in the diversity association by race versus gender as shown in Figs 17 and 11 were curious. While the race data is arguably less good than the gender, race appears to have a more powerful effect, which deserves further discussion.

Addressed: We thank the reviewer for this thoughtful suggestion. We have added more discussion of the field effects. In gender diversity, social science fields such as psychology and sociology have stronger gender diversity association effects. The effect is smaller for arts and humanities, especially history. We find less variance in racial diversity association effects across fields. Some STEM fields such as materials science and physics have relatively higher racial diversity association effects than social science fields like political science.

We find a significant diversity association effect in both gender and race in scientific collaboration networks, and the racial effect appears to be stronger than the gender effect. Gender diversity has strong fluctuation by fields, whereas racial diversity is more profoundly influenced by the demographics of residing countries. Exogenous factors such as time, institutional prestige, and group size that are commonly believed to have strong effects on workforce composition and researcher performance, tend to have little or no consistent effect in diversity socialization.

In the revised text, we write the following: “ *We observe a strong diversity association effect in both gender and race in scientific collaboration networks. The external factors that*

drive these dynamics may have different socio-economic mechanisms. Gender diversity has strong fluctuation by fields, and in particular, social science fields like psychology and sociology exhibit stronger gender diversity association effects (Supplementary Figure 8). On the other hand, the effect is weaker in arts & humanities fields, especially history. Although some STEM fields, such as materials science and physics, have relatively higher racial diversity association effects than social science fields like political science, there appears to be less variance in terms of racial diversity association effects across fields (Supplementary Figure 21). Racial diversity is also strongly influenced by the demographics of each particular country. The name-based gender/race labeling approach we employed may also have an impact on some effect sizes. The majority of researchers in all major racial groups can be assigned a racial distribution based on last names, while most Asian researchers cannot obtain accurate gender labeling with first names. This may give the racial analysis a more complete mapping of diversity socialization processes in academia. The large coefficients on the subfield and country variables in the regression analyses indicate that demographic trends overall drive the majority of the diversification effect, with early group diversity playing a substantial and significant secondary role in further increasing group diversity in the established period. "

As the researchers call for future research, they might suggest future qualitative work that works through the processes that underpin diversity association effects.

Addressed: We agree with the Reviewer and have provided more discussion of potential directions for future work in the Discussion section of the main text: *"More research is also needed to characterize other exogenous factors that may influence diversity association effects in science. For example, how do the researchers' diversity preferences vary across institutions with different prestige levels? To what degree is diversity socialization related to culture, economic development, or scientific infrastructure disparities across countries? Why does diversity socialization vary across fields? In addition to future quantitative work, careful qualitative work is needed to identify the specific social processes and choices that underpin the diversity associations we quantify."*

Generally, a strong paper, hope my comments are helpful.

Addressed: We thank the Reviewer again for the insightful comments. Based on these suggestions, we have included several additional analyses and discussions in the main manuscript and the Supplementary Information. We feel we have substantially improved the paper.

Response to Reviewer 2

The manuscript, "Gender and Racial Diversity Socialization in Science," investigates how early-career experiences with gender and racial diversity in mentor-mentee networks and collaboration networks influence the diversity of researchers' collaboration networks as they advance in their careers. It focuses on understanding the "diversity association effect," wherein early exposure to diverse environments may shape researchers' long-term preferences and behaviors regarding diversity. The study uses two datasets for analysis: (1) a mentor-mentee dataset covering approximately 17,900 researchers and (2) a co-authorship dataset encompassing over 560,000 scholars. A computational null model is proposed to support the authors' claims, controlling for confounding factors such as subfield, country, and institutional prestige.

The question I have about the manuscript are quite basic. Many of them reside in data and empirical analysis.

Addressed: We thank the Reviewer for the careful reading of our manuscript and for the insightful suggestions to improve it. We have included several additional analyses and discussions in the main manuscript and the Supplementary Information, and modified several figures and typos. We believe that these changes have substantially improved the readability of our paper and the robustness of our analyses.

The data description:

Several aspects of the data remain unclear after reviewing the manuscript and Supplementary Information (SI). For instance, the authors did not describe the proportion of researchers without complete first names. Are those researchers with only first initials significant enough to potentially alter the main findings? Additionally, it would be helpful to include basic descriptive statistics in the SI, such as the gender distribution across the 19 fields and across different cohorts (e.g., based on the year of a scholar's first publication). These details would provide readers with better context for understanding the data.

Addressed: We adopt a name-based gender assignment method to infer gender of researchers. We use the first names of authors from the publication data and match them with the gender association obtained from demographic census data²¹. This method does not apply to authors that use only initials of first names, which we show in Fig. R3a. Physicists have the highest preference of the initial letter usage in first names, and only about half of them write full names in their work. In natural science and engineering fields, about one-fourth of researchers use name initials. Social scientists generally have the lowest ratio of name initial usage. Fields with less frequent use of name initials have a higher proportion of

authors eligible for the name-based gender assignment procedure. We clarify these points in the revised manuscript.

Among the researchers that we assign gender, we show the proportion of women by field in Fig. R3b. Researchers in humanities and social science fields have the highest women ratio, especially in arts, psychology, and sociology. Some natural science fields, such as biology and medicine, also have a more gender-balanced workforce, with about half of researchers being women. Computer science, engineering, mathematics, and physics are among the fields with the lowest women percentage, with only one out of four researchers being women.

The proportion of women researchers has been increasing over the past five decades (Fig. R3c). However, the speed of growth over time is heterogeneous across fields, with some arts and humanities and social science fields having higher growth rates, while some engineering fields such as computer science have more moderate growth rates of women researchers. Some fields, such as biology, have reached a gender balance that has remained stable over the past decade or two.

I found the use of shapes and colors in Figure 1 to represent gender and the focal author somewhat confusing. The color pair assigned to men and women feels counterintuitive. Many other studies I have read use the opposite color combination. Revisiting the visualization design may enhance clarity.

Addressed: The Reviewer made a good suggestion here, and we agree that switching the color pair would be a more intuitive color combination. We have changed the color pair assigned to men and women in Figs. 1 of the main paper. Now, the male focal author is in green, and the female focal author is in purple (Fig. R4). Since we are consistent in our color scheme across all the figures in the main text and the Supplementary Information, we also adjust Figs. 2-3 in the main text accordingly (see Fig. R5 and Fig. R6) and the plots for gender diversity in the mentorship and coauthorship networks in the Supplementary Information.

The empirical analysis/mechanism:

First, there appear to be some missing controls in the null model. For instance, while the model considers pairs active in the same year, it might also need to account for cohort effects (i.e., the year an author published their first paper). Additionally, the cumulative citations of

Figure R3: **Descriptive statistics of the author names and gender composition.** **a**, proportion of authors with name initials in the publication data. **b**, proportion of women among authors that are assigned gender using their first names. **c**, time dynamics of the proportion of women among researchers based on the first year of publication.

the senior author at the time of link formation could be a valuable factor to consider when rewiring links.

Addressed: For the gender diversity analysis in the coauthorship network, we conduct a robustness test by incorporating two additional control variables into the null model, which is the first publishing year of authors and the career cumulative citation counts of senior authors. We divide researchers into four tiers based on their total citation counts up to the year of

Figure R4: Figure 1 in the main paper.

Figure R5: Figure 2 in the main paper.

the collaboration, with the first tier being authors receiving over 1,000 citations, the second tier being authors who have between 300 and 1,000 citations, the third tier being authors that have received less than 300 but more than 100 citations, and the rest being the fourth tier. In the new null models, we find that the gender diversity association effect persists for established researchers in both the aggregate analysis and the time dynamics (Fig. R7).

We conduct another robustness test by incorporating two additional control variables into the null model, which is the first publishing year of authors and the career cumulative

Figure R6: Figure 3 in the main paper.

citation counts of senior authors. After running the new null models, we find that the racial diversity association effect persists for established researchers in both the aggregate analysis and the time dynamics (Fig. R8).

We appreciate the suggestion of the Reviewer and agree that these could be valuable control variables to investigate. To that end, we began by adding them to the robustness analyses in order to quickly assess whether their inclusion qualitatively alters our findings. The results in these revised robustness tests indicate that including these additional variables does not change the results. This leads us to expect that including them in the null model (which would require a more significant refactoring of the model specification due to its network nature, and then re-running all the data analytics) would not change our main results. To help the reader understand this point, the regression models we report now in the main text include these variables and a brief discussion of them.

Secondly, some results in the regression analysis (e.g., SI Table 1 and Table 2) are unclear. For example, why do some variables have zero coefficients and zero standard errors? If these

Figure R7: **Robustness tests by inserting first publishing year and senior author citation counts in the null model.** We regard the established period as the years after the 6th publishing career year of a researcher. **a**, the gender diversity association effect persists for established researchers. **b**, the gender diversity association effect over time is similar to that in the main text. (** $p < 0.01$; *** $p < 0.001$; * $p < 0.05$).

are fixed-effect variables, their coefficients should not be reported in the table. Clarifying this issue would help avoid confusion.

Addressed: We thank the Reviewer for the careful reading of the regression tables, and we agree that it is necessary to avoid these zero coefficients and standard errors. We have added a coefficient to the original definition of institutional prestige,

$$p_i^{\text{inst}} = \frac{1}{100} \frac{N_i^{\text{high}} - \langle N^{\text{high}} \rangle}{\sigma / \sqrt{n^{\text{inst}}}}, \quad (1)$$

where N_i^{high} is the number of highly cited papers produced by institution i , $\langle N^{\text{high}} \rangle$ is the average number of highly cited papers by all institutions, σ is the standard deviation of highly cited papers, and n^{inst} is the number of institutions in the subfield. The coefficient $\frac{1}{100}$ is to ensure that institutional prestige has a moderate coefficient value in the regression models. In the new regression tables, the awkward 0.000 coefficients have been avoided (see Table R3 as an example).

Thirdly, there are a few typos in the SI. For instance, in the section describing the null model for co-authorship networks, page 19 states: "We repeat this process for all advisor-advisee pairs for each iteration of the network null model randomization." This appears to be a direct

Table R3: Linear regression models to predict the proportion of women among researchers' advisees.

Dependent variable: Model:	Women (%) among researchers' advisees			
	(1)	(2)	(3)	(4)
(Intercept)	0.329*** (0.018)	0.304*** (0.018)	0.039* (0.019)	-0.181*** (0.031)
Institutional prestige	-0.018 (0.019)	-0.019 (0.019)	-0.038* (0.018)	-0.037* (0.018)
Advisor group size	0.000 (0.000)	-0.000 (0.000)	0.001 (0.000)	0.001 (0.000)
Researcher is woman	0.194*** (0.007)	0.087*** (0.009)	0.084*** (0.009)	0.083*** (0.009)
Women(%) by subfield			0.845*** (0.024)	0.830*** (0.024)
Women(%) by country				0.637*** (0.070)
Women(%) in advisor's group		0.216*** (0.012)	0.122*** (0.012)	0.118*** (0.012)
R ²	0.053	0.071	0.140	0.145
Adj. R ²	0.053	0.071	0.140	0.144
Num. obs.	15637	15637	15637	15637

Robust standard-errors in parentheses

Signif. Codes: *** $p < 0.001$; ** $p < 0.01$; * $p < 0.05$

Figure R8: **Robustness tests by inserting first publishing year and senior author citation counts in the null model.** We regard the established period as the years after the 6th publishing career year of a researcher. **a**, the racial diversity association effect persists for established researchers. **b**, the racial diversity association effect over time is similar to that in the main text. (** $p < 0.001$; * $p < 0.01$; * $p < 0.05$).

copy-paste from SI section 1.2 and should be revised. Please ensure the manuscript and SI are carefully copy-edited.

Addressed: We have corrected this typo in the text: “We repeat this process for all coauthor pairs for each iteration of the network null model randomization.” We also checked the other sections of the paper and SI to ensure that they are carefully copy-edited.

Fourth, while the paper effectively demonstrates how early-career experiences with gender and racial diversity influence long-term collaboration behaviors, it could also address a critical question: do these preferences benefit science? For instance, do such collaborations lead to high-impact and novel scientific discoveries? Investigating these outcomes could further strengthen the paper’s contribution.

Addressed: The Reviewer has raised an insightful point here about what other scientific outcomes can be related to diversity socialization, such as the impact of articles produced by specific collaborative pairings of researchers. We feel that these investigations may need other carefully designed research projects to avoid bias and discrimination of certain cohorts of researchers, as endorsing “meritocratic” outcomes of certain collaborations may also simultaneously discredit the work of other groups that may not appear to be as equally productive or impactful. For instance, a paper published in 2020 that examines the gendered pairing of coauthors and research impact has aroused broad controversy and criticisms in social media, especially among women researchers²².

Therefore, while we find these recommendations valuable and interesting, we believe exploring these new research questions would be more suitable for future work. There is substantial qualitative evidence in the literature that more diverse teams produce better and more impactful science, which we include in the Introduction of the main paper: *“Hence, a clearer understanding of how different experiences of academic socialization influence downstream choices by researchers would connect evidence of the scientific and social advantages of diverse teams²³⁻²⁷ with potential policy or cultural interventions to accelerate the benefits to science and society.”*

I hope these comments are helpful in the development of your manuscript.

Addressed: We thank the Reviewer for these valuable and thoughtful comments. Addressing them has substantially improved the manuscript.

Response to Reviewer 3

This manuscript uses a null model to explore the “diversity association” between a researcher’s mentorship and collaboration during their training and establishment phases, with a focus on gender and racial diversity. The rewire method is well-established in the field of physics, and the results are statistically robust and convincing, and presented in a clear form.

Addressed: Thank you for your praise of our work and providing valuable comments to improve our manuscript. Addressing these comments has enhanced the quality of our manuscript, and led us to think about the project from new perspectives.

I have one minor suggestion. While the authors have provided qualitative conclusion supported by strong evidence regarding the presence of a diversity association, I wonder if they could expand the analysis to quantify this effect. Specifically, could the authors explore the expected impact of a 1% increase in gender/racial diversity during the training phase on the diversity observed in the establishment phase? Is the response linear, or does it exhibit a saturation point at higher levels of diversity?

Addressed: The Reviewer has raised a great question that expanded the scope of our analysis in a meaningful way. In the gender analysis on the coauthorship network, we use $\Delta_{\text{early}}^{\text{gender}} = \rho_{\text{early}} - \langle \rho_{\text{early}}^{\text{null}} \rangle$ to denote the difference between the gender diversity of a researcher’s early coauthors relative to the null model. We examine the diversity socialization effects $\Delta_{\text{established}}^{\text{gender}}$ in the established period as a function of gender diversity $\Delta_{\text{early}}^{\text{gender}}$ among early career coauthors. We find that women have an average $\Delta_{\text{early}}^{\text{gender}} \simeq 0$, suggesting that women tend to have a cohort of early coauthors close to the expected level of gender diversity (Fig. R9a). In contrast, men’s early coauthors have an average $\Delta_{\text{early}}^{\text{gender}} \simeq -0.2$, which is below the expected gender diversity in the environment. While we have demonstrated the existence of gender diversity association effect in the coauthorship network, which is $P(\Delta_{\text{established}}^{\text{gender}} > 0 | \Delta_{\text{early}}^{\text{gender}} > 0) > P(\Delta_{\text{established}}^{\text{gender}} > 0 | \Delta_{\text{early}}^{\text{gender}} \leq 0)$, the association between $\Delta_{\text{established}}^{\text{gender}}$ and $\Delta_{\text{early}}^{\text{gender}}$ is complicated (Fig. R9b). When $-0.2 < \Delta_{\text{early}}^{\text{gender}} < 0.5$, increasing $\Delta_{\text{early}}^{\text{gender}}$ leads to a positive growth of $\Delta_{\text{established}}^{\text{gender}}$. In other ranges, however, $\Delta_{\text{early}}^{\text{gender}}$ and $\Delta_{\text{established}}^{\text{gender}}$ appears to have a negative correlation.

In the racial analysis, we use $\Delta_{\text{early}}^{\text{race}} = \rho_{\text{early}} - \langle \rho_{\text{early}}^{\text{null}} \rangle$ to denote the difference between the racial diversity of a researcher’s early coauthors relative to the null model. We examine the diversity socialization effects $\Delta_{\text{established}}^{\text{race}}$ in the established period as a function of racial diversity $\Delta_{\text{early}}^{\text{race}}$ among early career coauthors. We find that researchers have an average $\Delta_{\text{early}}^{\text{race}} \simeq -0.2$, suggesting that researchers tend to have a cohort of early coauthors below

Figure R9: **Gender diversity socialization in the early and established period.** We use Δ_{early}^{gender} to denote the difference between the gender diversity of a researcher's early coauthors relative to the null model. **a**, the number of researchers and early career gender diversity of coauthors. **b**, associating the gender diversity socialization in the early and established period of individual researchers. Shaded areas represent 95% confidence intervals.

the expected level of racial diversity (Fig. R10a). The racial diversity of coauthors in the established period $\Delta_{established}^{race}$ is monotonically increasing as a function of the racial diversity of early coauthor Δ_{early}^{race} (Fig. R10b).

Additionally, there is a typo: I believe "racial" in Line 315 should be "gender."

Addressed: Thank you for the careful read-through of our paper. We have modified this typo in the manuscript: *"We use logistic regression models to examine how having a high percentage of women coauthors in the early career of researchers predicts the outcome variable, which is a binary coding of whether the gender diversity of junior coauthors is high relative to the null model when they become established researchers."*

Overall, this is a well-prepared and thoughtfully presented manuscript, and I look forward to seeing it published.

Addressed: Thank you for the recognition of our manuscript and the points raised to improve it. We hope that we have amended the paper to your full satisfaction.

Figure R10: **Racial diversity socialization in the early and established period.** We use Δ_{early}^{race} to denote the difference between the racial diversity of a researcher’s early coauthors relative to the null model. **a**, the number of researchers and early career racial diversity of coauthors. **b**, associating the racial diversity socialization in the early and established period of individual researchers. Shaded areas represent 95% confidence intervals.

(Remarks on code availability): The code is accessible. It would be helpful if the authors provide a readme file.

Addressed: We have prepared a README file in the Code Ocean platform where we deposit our code, and we also attach a copy to the end of this response memo (see APPENDIX I). The README file helps the reader understand the background and functionality of the null model, and the data and software necessary to reproduce and replicate the models.

References

1. Mortimer, J. T. & Simmons, R. G. Adult socialization. *Annu. Rev. Sociol.* **4**, 421–454 (1978).
2. Coleman, J. S. Social capital in the creation of human capital. *Am. J. Sociol.* **94**, S95–S120 (1988).
3. Austin, A. E. Preparing the next generation of faculty: Graduate school as socialization to the academic career. *J. High. Educ.* **73**, 94–122 (2002).
4. Marsden, P. V. The sociology of James S. Coleman. *Annu. Rev. Sociol.* **31**, 1–24 (2005).
5. Guhin, J., Calarco, J. M. & Miller-Idriss, C. Whatever happened to socialization? *Annu. Rev. Sociol.* **47**, 109–129 (2021).
6. Maher, M. A., Gilmore, J. A., Feldon, D. F. & Davis, T. E. Cognitive apprenticeship and the supervision of science and engineering research assistants. *J. Res. Pract.* **9**, M5 (2013).
7. Brunsma, D. L., Embrick, D. G. & Shin, J. H. Graduate students of color: Race, racism, and mentoring in the white waters of academia. *Sociol. Race Ethn.* **3**, 1–13 (2017).
8. Baker, V. L. & Griffin, K. A. Beyond mentoring and advising: Toward understanding the role of faculty “developers” in student success. *About Campus* **14**, 2–8 (2010).
9. Spoon, K., LaBerge, N., Wapman, K. H., Zhang, S., Morgan, A. C., Galesic, M., Fosdick, B. K., Larremore, D. B. & Clauset, A. Gender and retention patterns among US faculty. *Sci. Adv.* **9**, eadi2205 (2023).
10. Casad, B. J., Franks, J. E., Garasky, C. E., Kittleman, M. M., Roesler, A. C., Hall, D. Y. & Petzel, Z. W. Gender inequality in academia: Problems and solutions for women faculty in STEM. *J. Neurosci. Res.* **99**, 13–23 (2021).
11. Clauset, A., Arbesman, S. & Larremore, D. B. Systematic inequality and hierarchy in faculty hiring networks. *Sci. Adv.* **1**, e1400005 (2015).
12. Morgan, A. C., Way, S. F., Hoefler, M. J., Larremore, D. B., Galesic, M. & Clauset, A. The unequal impact of parenthood in academia. *Sci. Adv.* **7**, eabd1996 (2021).
13. Budig, M. J. & England, P. The wage penalty for motherhood. *Am. Sociol. Rev.* 204–225 (2001).

14. England, P., Bearak, J., Budig, M. J. & Hodges, M. J. Do highly paid, highly skilled women experience the largest motherhood penalty? *Am. Sociol. Rev.* **81**, 1161–1189 (2016).
15. Killewald, A. A reconsideration of the fatherhood premium: Marriage, coresidence, biology, and fathers' wages. *Am. Sociol. Rev.* **78**, 96–116 (2013).
16. England, P. The gender revolution: Uneven and stalled. *Gender Soc.* **24**, 149–166 (2010).
17. Masters-Waage, T., Spitzmueller, C., Edema-Sillo, E., St. Aubin, A., Penn-Marshall, M., Henderson, E., Lindner, P., Werner, C., Rizzuto, T. & Madera, J. Underrepresented minority faculty in the usa face a double standard in promotion and tenure decisions. *Nat. Hum. Behav.* 1–12 (2024).
18. Zhang, S., Wapman, K. H., Larremore, D. B. & Clauset, A. Labor advantages drive the greater productivity of faculty at elite universities. *Sci. Adv.* **8**, eabq7056 (2022).
19. Wapman, K. H., Zhang, S., Clauset, A. & Larremore, D. B. Quantifying hierarchy and dynamics in US faculty hiring and retention. *Nature* **610**, 120–127 (2022).
20. Leslie, S.-J., Cimpian, A., Meyer, M. & Freeland, E. Expectations of brilliance underlie gender distributions across academic disciplines. *Science* **347**, 262–265 (2015).
21. US Census Bureau, Frequently occurring surnames from the 2010 Census. https://www.census.gov/topics/population/genealogy/data/2010_surnames.html (2010). Accessed: 2022-09-30.
22. AlShebli, B., Makovi, K. & Rahwan, T. Retracted article: The association between early career informal mentorship in academic collaborations and junior author performance. *Nat. Commun.* **11**, 1–8 (2020).
23. Page, S. E. *The Difference: How the Power of Diversity Creates Better Groups, Firms, Schools, and Societies*. (Princeton University Press, 2008).
24. Hofstra, B., Kulkarni, V. V., Munoz-Najar Galvez, S., He, B., Jurafsky, D. & McFarland, D. A. The diversity–innovation paradox in science. *Proc. Natl. Acad. Sci. USA* **117**, 9284–9291 (2020).
25. Yang, Y., Tian, T. Y., Woodruff, T. K., Jones, B. F. & Uzzi, B. Gender-diverse teams produce more novel and higher-impact scientific ideas. *Proc. Natl. Acad. Sci. USA* **119**, e2200841119 (2022).
26. Uzzi, B., Mukherjee, S., Stringer, M. & Jones, B. Atypical combinations and scientific impact. *Science* **342**, 468–472 (2013).

27. Nielsen, M. W., Alegria, S., Börjeson, L., Etzkowitz, H., Falk-Krzesinski, H. J., Joshi, A., Leahey, E., Smith-Doerr, L., Woolley, A. W. & Schiebinger, L. Gender diversity leads to better science. *Proc. Natl. Acad. Sci. USA* **114**, 1740–1742 (2017).

APPENDIX I

Gender and racial diversity socialization in science

The gender and racial composition of the scientific workforce has never been representative of the population in almost every major nation in the world. Recent research has examined the gender and racial inequality in science in many aspects, including the enrollment of graduate and doctoral training programs, faculty hiring in universities, funding support, publishing in prestigious journals, and more. The prevalence of these disparities and biases in science has led many scholars to advocate for more equal opportunities in the scientific community, for women, ethnic minority people, and other under-represented social groups. Yet, it remains unclear if there are plausible methods or treatments that may help mitigate the existing inequalities in science.

Researchers form collaboration networks to share ideas, transfer academic knowledge and skills, and work collectively to tackle challenging scientific problems. Since researchers from different demographic cohorts may have varied preferences in building connections, and collaboration networks may also play an important role in shaping values towards diversity and inclusion among members. In our work, we assume that having worked in a gender/racially diverse environment would socialize researchers in a way that could promote their awareness and willingness towards diversity and inclusion, and increase the chances that these researchers nurture gender/racially diverse research groups when they become established researchers.

We introduce two null models to estimate the level of gender/racial diversity under a fully randomized scenario. The gender/racial composition of researchers may depend largely on a number of social factors, including time, subfield, country, and institutional prestige. In the null model setting, we first construct the mentorship/coauthorship network, then we randomly rewire the links by controlling these social factors. After a large enough number of rewiring attempts, we take the randomized network as one replication sample in the null model. We repeat this process to assemble 100 randomized network samples and compute the average value of gender/racial diversity of individual researchers, and use this quantity as the baseline diversity.

Networks and null models

We introduce null models for the mentorship network and coauthorship network. These two networks are slightly different in the data format and network construction processes. In the mentorship data, an advisor would supervise an advisee for a few years, which we show in the figure below. To rewire the mentorship network, for a specific year, we randomly select another advisee from the network and check if it satisfies the control conditions of subfield, country, and institutional prestige. If the conditions are met, we reshuffle the two advisees in the mentorship network. We repeat this process for a large enough number of iterations to obtain a randomized mentorship network.

The coauthorship networks are constructed using publication data. A group of researchers are regarded as coauthors if their names appear in one paper. For this analysis we exclude all single-author papers. To rewire the coauthorship network, we randomly select another researcher from the network and check if it satisfies the control conditions of publishing year, subfield, country, and institutional prestige. If the conditions are met, we reshuffle the two researchers in the coauthorship network. We repeat this process for a large enough number of iterations to obtain a randomized coauthorship network. Details of the rewiring process can be found in the illustrative figure below.

Data

We provide details of the mentorship and coauthorship data that are used to run the diversity socialization null models.

For the mentorship data:

The mentorship.txt file contains four columns: mentorId, menteed, firstYr, and lastYr. MentorId and menteed are author IDs for the advisors and advisees, and firstYr refers to the first year that the focal advisor-advisee pair started, and lastYr depicts the year this pair ended.

The mentorship_aut_country.txt file contains two columns: AuthorId, and country. AuthorId shares the same ID code with the mentorId and menteed in the mentorship.txt data-set. This file provides information of countries for researchers in the mentorship data.

The mentorship_aut_institution_rank.txt file contains two columns: AuthorId, and institution_rank. This file provides information of the rank of institutions for researchers in the mentorship data. Institutions are ranked in four tiers, the top tier being the upper 1th percentile, the second tier being the upper 2-10th percentile, the third tier being the upper 11-20th percentile, and the remainder being the fourth tier.

The mentorship_aut_subfield.txt file contains two columns: AuthorId, FieldOfStudyId. This file provides information of the subfields for researchers in the mentorship data.

For the coauthorship data:

The coauthorship.txt file contains four columns: PaperId, AuthorId, AffiliationId, and Year. It contains the paper ID, author Id, the author affiliations, and the year of publication. Authors who have worked on the same paper are regarded as coauthors, and this data file is used to construct the coauthorship network.

The coauthorship_papAut_country.txt file contains three columns: PaperId, AuthorId, and country. This file provides information of countries for researchers in the coauthorship data.

The coauthorship_papAut_institution_rank.txt file contains three columns: PaperId, AuthorId, and institution_rank. This file provides information of institutional prestige rank for researchers in the coauthorship data. The ranks are defined the same way as in the mentorship data.

The coauthorship_papAut_subfield.txtfile contains three columns: PaperId, AuthorId, and FieldOfStudyId. This file provides information of the subfields for researchers in the coauthorship data.

Software

```
R version 4.3.3 "Angel Food Cake" (2024-02-29)
data.table version 1.15.4
dplyr version 1.1.4
bit64 version 4.0.5
```

LICENSE

MIT License

Copyright (c) School of Artificial Intelligence, Beihang University. All rights reserved.

Permission is hereby granted, free of charge, to any person obtaining a copy of this software and associated documentation files (the "Software"), to deal in the Software without restriction, including without limitation the rights to use, copy, modify, merge, publish, distribute, sublicense, and/or sell copies of the Software, and to permit persons to whom the Software is furnished to do so, subject to the following conditions:

The above copyright notice and this permission notice shall be included in all copies or substantial portions of the Software.

THE SOFTWARE IS PROVIDED "AS IS", WITHOUT WARRANTY OF ANY KIND, EXPRESS OR IMPLIED, INCLUDING BUT NOT LIMITED TO THE WARRANTIES OF MERCHANTABILITY, FITNESS FOR A PARTICULAR PURPOSE AND NONINFRINGEMENT. IN NO EVENT SHALL THE AUTHORS OR COPYRIGHT HOLDERS BE LIABLE FOR ANY CLAIM, DAMAGES OR OTHER LIABILITY, WHETHER IN AN ACTION OF CONTRACT, TORT OR OTHERWISE, ARISING FROM, OUT OF OR IN CONNECTION WITH THE SOFTWARE OR THE USE OR OTHER DEALINGS IN THE SOFTWARE.

Response to the reviews of manuscript NATCOMPUTSCI-24-2778A: “Gender and racial diversity socialization in science”

Dear Editors and Reviewers,

Thank you for your detailed attention to our manuscript and for providing constructive comments in the peer review process. We have addressed the final request of Reviewer 2 in this response letter. We feel that the revised manuscript has improved substantially.

Our responses are organized into the following sections:

- Response to the Editor
- Response to Reviewer 1
- Response to Reviewer 2
- Response to Reviewer 3

Sincerely,

Weihua Li, Hongwei Zheng, Jennie E. Brand & Aaron Clauset

Response to the Editor

Thank you for submitting your revised manuscript "Gender and racial diversity socialization in science" (NATCOMPUTSCI-24-2778A). It has now been seen by the original referees and their comments are below. The reviewers find that the paper has improved in revision, and therefore we'll be happy in principle to publish it in Nature Computational Science, pending minor revisions to satisfy the referees' final requests and to comply with our editorial and formatting guidelines.

Addressed: We thank the Editor for the opportunity to revise our manuscript for final submission. We have addressed the last comment of Reviewer 2 the Supplementary Information and reported the details in this response letter. We have also revised our manuscript to meet the editorial requests and formatting guidelines, and prepared the high-resolution figures and other files needed for publication.

Response to Reviewer 1

The authors have systematically revised the paper, addressing the comments of the reviewers thoughtfully. As a result, what was a strong paper, is stronger now.

I appreciate the careful attention to detail, and the important contributions the paper makes.

Addressed: Thank you for the positive endorsement of our revision and all the constructive comments, which are critically helpful to improve our manuscript.

Response to Reviewer 2

Thank you for addressing my previous comments. I am satisfied with the current revised version. However, one minor issue may require your attention: the selection of thresholds for classifying author cumulative citations. Specifically, the rationale behind choosing 100, 300, and 1,000 as thresholds needs further clarification. Are these values based on the 25th, 50th, and 75th percentiles of the distribution? Providing a justification for these cutoffs would strengthen the methodological transparency of your analysis.

Addressed: We thank the Reviewer for the insightful suggestions to improve our manuscript in the peer review process. Regarding the remaining comment, we have carefully explained the rationale behind using these numbers to divide established researchers in the Supplementary Information.

Citation counts are often used to assess the impact of researchers, yet this metric varies substantially over time and across fields. For example, citation counts received by a recently published paper can be times more than a paper published half a century ago. The number of citations a materials science paper garners is also substantially larger than a paper on pure mathematics. Since our standard of citation impact should be reasonable and applicable across several decades from 1970 to 2020, and for all subfields in arts and humanities, natural sciences, mathematics, engineering, and social sciences, we use an approximation of 100 citations as a lower bar of research impact and 1,000 citations as an upper level of impact for individual researchers. As such, we divide researchers into four tiers based on their total citation counts up to the year of the collaboration, with the first tier being authors receiving over 1,000 citations, the second tier being authors who have between 300 and 1,000 citations, the third tier being authors that have received less than 300 but more than 100 citations, and the rest being the fourth tier.

Response to Reviewer 3

The authors have cleared all my suggestions and concerns. I'll be happy to see this manuscript on Nature Computational Science.

Junming Huang

Addressed: Thank you for your endorsement of our work and for providing suggestions to improve our manuscript.